# Noise Consistency Training: A Native Approach for One-Step Generator in Learning Additional Controls

**Yihong Luo**[1][*] **Shuchen Xue**[2][*] **Tianyang Hu**[3][†] **Jing Tang**[4,1][†]
[1]HKUST [2]UCAS [3]CUHK(SZ) [4]HKUST(GZ)

## Abstract

The pursuit of efficient and controllable high-quality content generation remains a central challenge in artificial intelligence-generated content (AIGC). While one-step generators, enabled by diffusion distillation techniques, offer excellent generation quality and computational efficiency, adapting them to new control conditions—such as structural constraints, semantic guidelines, or external inputs—poses a significant challenge. Conventional approaches often necessitate computationally expensive modifications to the base model and subsequent diffusion distillation. This paper introduces Noise Consistency Training (NCT), a novel and lightweight approach to directly integrate new control signals into pre-trained one-step generators without requiring access to original training images or retraining the base diffusion model. NCT operates by introducing an adapter module and employs a *noise consistency loss* in the noise space of the generator. This loss aligns the adapted model's generation behavior across noises that are conditionally dependent to varying degrees, implicitly guiding it to adhere to the new control. Theoretically, this training objective can be understood as minimizing the distributional distance between the adapted generator and the conditional distribution induced by the new conditions. NCT is modular, data-efficient, and easily deployable, relying only on the pre-trained one-step generator and a control signal model. Extensive experiments demonstrate that NCT achieves state-of-the-art controllable generation in a single forward pass, surpassing existing multi-step and distillation-based methods in both generation quality and computational efficiency.

## 1 Introduction

The pursuit of high-quality, efficient, and controllable generation has become a central theme in the advancement of artificial intelligence-generated content (AIGC). The ability to create diverse and realistic content is crucial for a wide range of applications, from art and entertainment to scientific visualization and data augmentation. Recent breakthroughs in diffusion models and their distillation techniques have led to the development of highly capable one-step generators [1, 2, 3, 4, 5]. These models offer a compelling combination of generation quality and computational efficiency, significantly reducing the cost of content creation. Methods such as Consistency Training [6] and Inductive Moment Matching [7] have further expanded the landscape of native few-step or even one-step generative models, providing new tools and perspectives for efficient generation.

However, as AIGC applications continue to evolve, new scenarios are constantly emerging that demand models to adapt to novel conditions and controls. These conditions can take many forms, encompassing structural constraints (e.g., generating an image with specific edge arrangements), semantic guidelines (e.g., creating an image that adheres to a particular artistic style), and external

---

[*]Core contribution.

[†]Corresponding authors: Tianyang Hu and Jing Tang.

39th Conference on Neural Information Processing Systems (NeurIPS 2025).

factors such as user preferences or additional sensory inputs (e.g., generating an image based on a depth map). Integrating such controls effectively and efficiently is a critical challenge.

The conventional approach to incorporating controls into diffusion models often involves modifying the base model architecture and subsequently performing diffusion distillation to obtain a one-step student model [8]. This process, while effective, can be computationally expensive and time-intensive, requiring significant resources and development time. A more efficient alternative would be to extend the distillation pipeline to accommodate new controls directly, potentially bypassing the need for extensive retraining of the base diffusion model [9]. However, even extending the distillation pipeline can still be a heavy undertaking, adding complexity and computational overhead. Therefore, the question of how to directly endow one-step generators with new controls in a lightweight and efficient manner remains a significant challenge.

In this paper, we answer this question by proposing Noise Consistency Training (NCT) — a simple yet powerful approach that enables a pre-trained one-step generator to incorporate new conditioning signals without requiring access to training images or retraining the base model. NCT achieves this by introducing an adapter module that operates in the noise space of the pre-trained generator. Specifically, we define a noise-space consistency loss that aligns the generation behavior of the adapted model across different noise levels, implicitly guiding it to satisfy the new control signal. Besides, we employ a boundary loss ensuring that when given a condition already associated with input noise, the generation should remain the same as one-step uncontrollable generation. This can ensure the distribution of the adapter generator remains in the image domain rather than collapsing. Theoretically, we demonstrate in Section 3.2 that this training objective can be understood as matching the adapted generator to the intractable conditional induced by a discriminative control model when the boundary loss is satisfied, effectively injecting the desired conditioning behavior.

Our method is highly modular, data-efficient, and easy to deploy, requiring only the pre-trained one-step generator and a control signal model, without the need for full-scale diffusion retraining or access to the original training data. Extensive experiments across various control scenarios demonstrate that NCT achieves state-of-the-art controllable generation in a single forward pass, outperforming existing multi-step and distillation-based methods in both quality and computational efficiency.

## 2 Preliminary

**Diffusion Models (DMs).** DMs [2, 1] operate via a forward diffusion process that incrementally adds Gaussian noise to data $\mathbf{x}$ over $T$ timesteps. This process is defined as $q(\mathbf{x}_t|\mathbf{x}) \triangleq \mathcal{N}(\mathbf{x}_t; \alpha_t \mathbf{x}, \sigma_t^2 \mathbf{I})$, where $\alpha_t$ and $\sigma_t$ are hyperparameters dictating the noise schedule. The diffused samples are obtained via $\mathbf{x}_t = \alpha_t \mathbf{x} + \sigma_t \epsilon$, with $\epsilon \sim \mathcal{N}(\mathbf{0}, \mathbf{I})$. The diffusion network, $\epsilon_\theta$ is trained by denoising: $\mathbb{E}_{\mathbf{x}, \epsilon, t}||\epsilon_\theta(\mathbf{x}_t, t) - \epsilon||_2^2$. Once trained, generating samples from DMs typically involves iteratively solving the corresponding diffusion stochastic differential equations (SDEs) or probability flow ordinary differential equations (PF-ODEs), a process that requires multiple evaluation steps.

**ControlNet.** Among other approaches for injecting conditions [10, 11, 12, 13, 14], ControlNet [15] has emerged as a prominent and effective technique for augmenting pre-trained DMs with additional conditional controls. Given a pre-trained diffusion model $\epsilon_\theta$, ControlNet introduces an auxiliary network, parameterized by $\phi$. This network is trained by minimizing a conditional denoising loss $L(\phi)$ to inject the desired controls:

$$L(\phi) = \mathbb{E}_{\mathbf{x}, \epsilon, t}||\epsilon - \epsilon_{\theta, \phi}(\mathbf{x}_t, \mathbf{c})||_2^2. \tag{1}$$

After training, ControlNet enables the integration of new controls into the pre-trained diffusion models.

**Maximum Mean Discrepancy.** Maximum Mean Discrepancy (MMD [16]) between distribution $p(\mathbf{x}), q(\mathbf{y})$ is an integral probability metric [17]:

$$\text{MMD}^2(p, q) = ||\mathbb{E}_{\mathbf{x}}[\psi(\mathbf{x})] - \mathbb{E}_{\mathbf{y}}[\psi(\mathbf{y})]||^2, \tag{2}$$

where $\psi(\cdot)$ is a kernel function.

**Diffusion Distillation.** While significant advancements have been made in training-free acceleration methods for DMs [18, 19, 20, 21, 22], diffusion distillation remains a key strategy for achieving high-quality generation in very few steps. Broadly, these distillation methods follow two primary paradigms: 1) Trajectory distillation [23, 24, 25, 26, 27, 28], which seeks to replicate the teacher

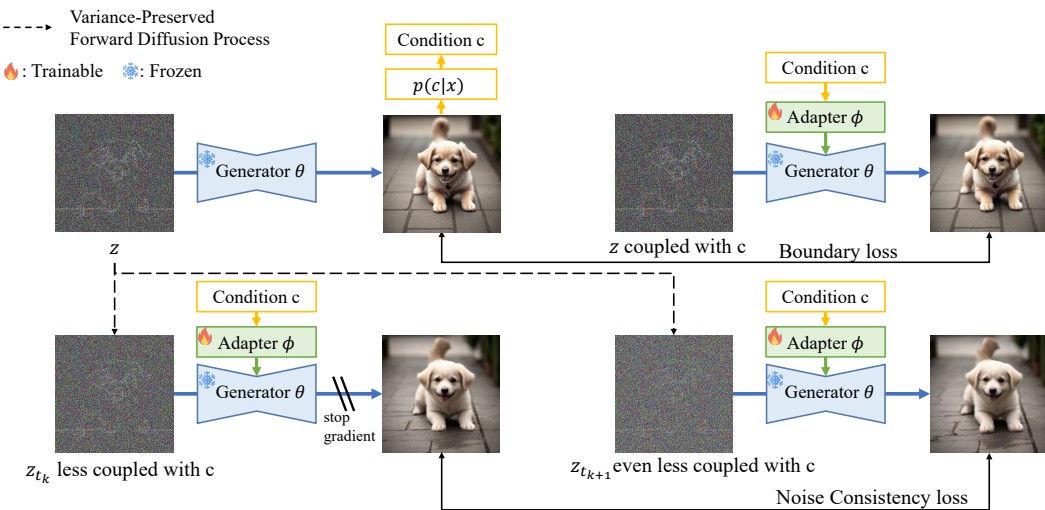

Figure 1: Framework description of our proposed **NCT**. We note that *we deliberately added some structural features to the noise to enhance readability*, rather than faithfully rendering Gaussian noise.

model's ODE trajectories on an instance-by-instance basis. These methods can encounter difficulties with precise instance-level matching. 2) Distribution matching, often realized via score distillation [5, 3, 29, 4], which aims to align the output distributions of the student and teacher models using divergence metrics. Our work utilizes a pre-trained one-step generator, which itself is a product of diffusion distillation; however, the training of our proposed NCT method does not inherently require diffusion distillation.

**Additional Controls for One-step Diffusion.** The distillation of multi-step DMs into one-step generators, particularly through score distillation, is an established research avenue [3, 5, 29]. However, the challenge of efficiently incorporating new controls into these pre-trained one-step generators is less explored. CCM [30], for example, integrates consistency training with ControlNet, demonstrating reasonable performance with four generation steps. In contrast, our work aims to surpass standard ControlNet performance in most cases using merely a single step. Many successful score distillation techniques [3, 5, 31, 32] rely on initializing the one-step student model with the weights of the teacher model. SDXS [8] explored learning controlled one-step generators via score distillation, but their framework requires both the teacher model and the generated "fake" scores to possess a ControlNet compatible with the specific condition being injected. JDM [9] minimizes a tractable upper bound of the joint KL divergence, which can teach a controllable student with an uncontrollable teacher. Generally, prior works are built on specific distillation techniques for adapting controls to one-step models. We argue that given an already proficient pre-trained one-step generator, performing an additional distillation for adding new controls is computationally expensive and unnecessary. However, *how to develop a **native** technique for one-step generators remains unexplored.* Our work takes the first step in designing a native approach for one-step generators to add new controls to one-step generators without requiring any diffusion distillation.

## 3 Method

**Problem Setup.** Let $\mathbf{z} \in \mathbb{R}^m$ be a latent variable following a standard Gaussian density $p(\mathbf{z})$. We have a pre-trained generator $f_\theta : \mathbb{R}^m \to \mathbb{R}^n$ that maps $\mathbf{z}$ to a data sample $\mathbf{x} = f_\theta(\mathbf{z})$. The distribution of these generated samples has a density $p_\theta(\mathbf{x})$, providing a high-quality approximation of the data distribution, such that $p_\theta(\mathbf{x}) \approx p_d(\mathbf{x})$. For any $\mathbf{x}$, there is a conditional probability density $p(\mathbf{c}|\mathbf{x})$ specifying the likelihood of condition c given $\mathbf{x}$. Our goal is to directly incorporate additional control c for a pre-trained one-step generator with additional trainable parameters $\phi$ (e.g. a ControlNet). More specifically, we aim to train a conditional generator $f_{\theta,\phi}(\mathbf{z}, \mathbf{c})$ that, when given a latent code $\mathbf{z}$ sampled from a standard Gaussian distribution and an independently sampled condition c, produces a sample $\mathbf{x}$ such that the joint distribution of $(\mathbf{x}, \mathbf{c})$ matches $p(\mathbf{x}, \mathbf{c}) = p_\theta(\mathbf{x})p(\mathbf{c}|\mathbf{x})$.

## 3.1 Failure modes of Naive Approaches for Adding Controls

Given a pre-trained diffusion model $\epsilon_\theta(\mathbf{x}_t, t)$, the adapters for injecting new conditions can be trained by minimizing a denoising loss [15, 10]. Hence, a natural idea for injecting new conditions into the pre-trained one-step generator is also adapting the denoising loss for training as follows:

$$\min_\phi d(f_{\theta,\phi}(\mathbf{z}, \mathbf{c}), \mathbf{x}), \ \mathbf{z} = \alpha_T \mathbf{x} + \sigma_T \epsilon, \ \mathbf{c} \sim p(\mathbf{c}|\mathbf{x}), \tag{3}$$

where $d(\cdot, \cdot)$ is a distance metric and $T$ denotes the terminal timestep. This approach can potentially inject new conditions into the one-step generator $f_\theta$, similar to existing adapter approaches for DMs. However, it fails to generate high-quality images — the resulting images are blurry, which is due to the *high variance of the optimized objective*. Specifically, its optimal solution is achieved at $f_{\theta,\phi}(\mathbf{z}, \mathbf{c}) = E[\mathbf{x}|\mathbf{z}, \mathbf{c}]$, which is an average of every potential image.

To reduce the variance, one may consider performing denoising loss over *coupled pairs* $(\mathbf{z}, \mathbf{x}, \mathbf{c})$, where $\mathbf{z} \sim \mathcal{N}(0, I)$, c is the condition corresponding to the generated samples $\mathbf{x} = f_\theta(\mathbf{z})$. However, such an approach is unable to perform conditional generation given random $\mathbf{z}$. This is because the model is only exposed to instances of $\mathbf{z}$ strongly associated with c (i.e., $\mathbf{c} \sim p(c|f_\theta(\mathbf{z}))$) during its training, and never encountered random pairings of c and $\mathbf{z}$.

High variance in denoising loss is also a key factor hindering fast sampling in diffusion models. Several methods have been proposed to accelerate the sampling of diffusion models, with optimization objectives typically characterized by low variance properties [26, 24, 33]. Among these, consistency models [26, 27] stand out as a promising approach — instead of optimizing direct denoising loss, they optimize the distance between denoising results of highly-noisy samples and lowly-noisy samples:

$$\min_\alpha L(\alpha) = d(g_\alpha(\mathbf{x}_{t_{n+1}}), \mathrm{sg}(g_\alpha(\mathbf{x}_{t_n}))), \tag{4}$$

where $\mathrm{sg}(\cdot)$ denotes the stop-gradient operator and $g_\alpha$ denotes the desired consistency models. Similar to denoising loss, consistency loss can also force networks to use conditions; thus, it can be used to train adapters to inject new conditions [30]. However, the consistency approach cannot be adapted to the one-step generator since it requires defining the loss over multiple noisy-level images, while the one-step generator only takes random noise as input.

## 3.2 Our Approach: Noise Consistency Training

To directly inject condition to one-step generator, we propose **Noise Consistency Training**, which diffuses noise to decouple it from the condition and operates the consistency training in **noise space**. Specifically, we diffuse an initial noise $\mathbf{z} \sim \mathcal{N}(0, I)$ to multiple levels $\mathbf{z}_t$ via variance-preservation diffusion as follows:

$$\mathbf{z}_t = \sqrt{1 - \sigma_t}\mathbf{z} + \sigma_t \epsilon, \tag{5}$$

where $\epsilon \sim \mathcal{N}(0, I)$. This ensures that $\mathbf{z}_t$ also follows the standard Gaussian distribution, thus it can be transformed to the high-quality image by the pre-trained one-step generator $f_\theta$.

To inject new conditions to $f_\theta$, we apply an adapter with parameter $\phi$, which transforms $f_\theta(\cdot)$ that only takes random noise as input to $f_{\theta,\phi}(\cdot, \cdot)$ that can take an additional condition c as input. We sample coupled pairs $(\mathbf{z}, \mathbf{c})$ from $p_\theta(\mathbf{z}, \mathbf{c})$, where $p(\mathbf{z}, \mathbf{c}) = p(\mathbf{z})p_\theta(\mathbf{c}|\mathbf{z})$, and $p_\theta(\mathbf{c}|\mathbf{z}) \triangleq p_\theta(\mathbf{c}|f_\theta(\mathbf{z}))$. By the $(\mathbf{z}, \mathbf{c})$ pairs, we can perform **Noise Consistency Loss** as follows:

$$\min_\phi \mathbb{E}_{p(\mathbf{z})p(\mathbf{c}|f_\theta(\mathbf{z}))} \mathbb{E}_{q(\mathbf{z}_{t_n}|\mathbf{z}), q(\mathbf{z}_{t_{n-1}}|\mathbf{z})} \mathbb{E}_{\epsilon \sim \mathcal{N}(0,I)} d(f_{\theta,\phi}(\mathbf{z}_{t_n}, \mathbf{c}), \mathrm{sg}(f_{\theta,\phi}(\mathbf{z}_{t_{n-1}}, \mathbf{c})))$$
$$= \mathbb{E}_{\mathbf{z}, \mathbf{c}|\mathbf{z}, \epsilon} d(f_{\theta,\phi}(\mathbf{z}_{t_n}, \mathbf{c}), \mathrm{sg}(f_{\theta,\phi}(\mathbf{z}_{t_{n-1}}, \mathbf{c}))), \quad \text{\#Simplified Notation} \tag{6}$$

where $\mathbf{z}_{t_n} = \sqrt{1 - \sigma_{t_n}^2}\mathbf{z} + \sigma_{t_n}\epsilon$ and $\mathbf{z}_{t_{n-1}} = \sqrt{1 - \sigma_{t_{n-1}}^2}\mathbf{z} + \sigma_{t_{n-1}}\epsilon$. The defined diffusion process can gradually diffuse the coupled pairs $(\mathbf{z}, \mathbf{c})$ to independent uncoupled pairs $(\mathbf{z}_T, \mathbf{c})$. By minimizing the distance between predictions given "less-coupled" pairs and "more-coupled" pairs, we can force the network to utilize the condition. Once trained, the consistency is ensured in the **noise space**. It is expected that the adapter $\phi$ can be trained for injecting new conditions c, while keeping the high-quality generation capability in one-step. Since the optimized objective has low variance and the generator $f_\theta$ can produce high-quality images, the adapter just need to learn how to adapt to the conditions c.

**Lemma 1.** *Define $p(\mathbf{z}_0|\mathbf{c}) \triangleq \frac{p(\mathbf{z}_0)p(\mathbf{c}|f_\theta(\mathbf{z}))}{p(\mathbf{c})}$ and $p(\mathbf{z}_t|\mathbf{c}) \triangleq \int q(z_t|\mathbf{z}_0)p(\mathbf{z}_0|\mathbf{c})dz_0$. The forward diffusion process defines an interpolation for the joint distribution $p(\mathbf{z}_t, \mathbf{c}) \triangleq p(\mathbf{z}_t|\mathbf{c})p(\mathbf{c})$ between $p(\mathbf{z}_0, \mathbf{c}) = p(\mathbf{c}|f_\theta(\mathbf{z}_0))p(\mathbf{z}_0)$ and $p(\mathbf{z}_T, \mathbf{c}) = p(\mathbf{z}_T)p(\mathbf{c})$.*

The above Lemma 1 provides a formal justification to our noise diffusion process as interpolation between the coupled pairs $(\mathbf{z}, \mathbf{c})$ to independent pairs $(\mathbf{z}_T, \mathbf{c})$. The proof can be found in Section A.

**Lemma 2.** *We define the $f_{\theta,\phi}(\sqrt{1 - \sigma_{t_{k+1}}^2}\mathbf{z} + \sigma_{t_{k+1}}\epsilon, \mathbf{c})$ induced distribution to be $p_{\theta,\phi,t_{k+1}}$. The proposed noise consistency loss is a practical estimation of the following loss:*

$$L(\phi) = \sum_{k=0}^{N-1} \mathrm{MMD}^2(p_{\theta,\phi,t_{k+1}}, p_{\theta,\phi,t_k}), \tag{7}$$

*under specific hyper-parameter choices (e.g., set particle samples to 1).*

See proof in the Section A. The above Lemma 2 builds the connection between our noise consistency training and conditional distribution matching. Technically speaking, using larger particle numbers can further reduce training variance. However, in practice, we found that directly using a single particle achieves similar performance and is more computationally feasible. More investigations on the effect of particle numbers can be found in Section B. This work serves as proof of concept that we can design an approach native to one-step generator in learning new controls, we leave other exploration for further reducing variance in future work.

**Boundary Loss** A core difference between NCT and CM lies in the model's behavior when reaching boundaries. Specifically, for CM, when the input reaches the boundary $\mathbf{x}_0$, the model only needs to degenerate into an identity mapping outputting $\mathbf{x}_0$, which can be easily satisfied through reparameterization g to stabilize the training. NCT, however, is fundamentally different — when the input reaches the boundary $\mathbf{z}$, the model cannot simply degenerate into an identity mapping, but needs to map $\mathbf{z}$ to high-quality clean images. This means this boundary is non-trivial — the network needs to learn to map $\mathbf{z}$ to corresponding images. Simply reparameterizing $f_{\theta,\phi}$ cannot fully stabilize the training. To satisfy this boundary condition and stabilize the training, we propose setting the clean image corresponding to $\mathbf{z}$ as $f_\theta(\mathbf{z})$ and implementing the following *boundary loss*:

$$\min_\phi \mathbb{E}_{\mathbf{z},\mathbf{c}|\mathbf{z},\epsilon} d(f_{\theta,\phi}(\mathbf{z}, \mathbf{c}), f_\theta(\mathbf{z})), \ \mathbf{z} \sim \mathcal{N}(0, I), \ \mathbf{c} \sim p(\mathbf{c}|f_\theta(\mathbf{z})). \tag{8}$$

By minimizing this loss, we can ensure the boundary conditions hold and constrain the generator's output to be close to the data distribution. Intuitively, this loss is easy to understand: when the generator receives the same noise $\mathbf{z}$ and conditions corresponding to $f_\theta$, its generation should be invariant. With the help of this loss, we can constrain the generator's output to stay near the data distribution — otherwise, if we only minimize the noise consistency loss, the model might find unwanted shortcut solutions.

**Theorem 1.** *Consider a parameter set $\phi$ that satisfies the following two conditions:*

*1. Boundary Condition: The parameters $\phi$ ensure the boundary loss is zero:*

$$\mathbb{E}_{p(\mathbf{z})p(\mathbf{c}|f_\theta(\mathbf{z}))}[d(f_{\theta,\phi}(\mathbf{z}, \mathbf{c}), f_\theta(\mathbf{z}))] = 0,$$

*2. Consistency Condition: The parameters $\phi$ also satisfy:*

$$L(\phi) = \sum_{k=0}^{N-1} \mathrm{MMD}^2(p_{\theta,\phi,t_{k+1}}, p_{\theta,\phi,t_k}) = 0$$

*Then $f_{\theta,\phi}$ maps independent $p(\mathbf{z})p(\mathbf{c})$ to the target joint distribution $p_\theta(\mathbf{x})p(\mathbf{c}|\mathbf{x})$.*

See proof in the Appendix. Theorem 1 provides theoretical insight for our optimization objective, which is an empirical version for practice.

**Overall Optimization** We observed that the noise consistency loss is only meaningful when boundary conditions are satisfied or nearly satisfied; otherwise, the generator $f_{\theta,\phi}$ can easily find undesirable shortcut solutions, thus we suggest using a constrained optimization form as follows:

**Algorithm 1** Noise Consistency Training

---

**Require:** Pre-trained One-Step Generator $f_\theta$, Adapter $\phi$, total iterations $N$
**Ensure:** Optimized adapter $\phi$ for injecting new condition.
1: **for** $i \leftarrow 1$ **to** $N$ **do**
2:      Sample noise $\mathbf{z}$ from standard Gaussian distribution;
3:      Sample noise $\epsilon$ from standard Gaussian distribution;
4:      Sample $\mathbf{x}$ with initialized noise $\mathbf{z}$ from frozen generator $f_\theta$, i.e., $\mathbf{x} = f_\theta(\mathbf{z})$.
5:      Sample condition c corresponding to $\mathbf{x}$ by $p(\mathbf{c}|\mathbf{x})$.
6:      # Primal Step:
7:      ## Diffuse Noise via Variance-Preserved Diffusion
8:      $z_{t_{k+1}} \leftarrow \alpha_{t_{k+1}}\mathbf{z} + \sigma_{t_{k+1}}\epsilon$ and $z_{t_k} \leftarrow \alpha_{t_k}\mathbf{z} + \sigma_{t_k}\epsilon$.
9:      ## Compute Noise Consistency Loss
10:     $\mathcal{L}_{con} \leftarrow d(f_{\theta,\phi}(\mathbf{z}_{t_{k+1}}, \mathbf{c}), \mathrm{sg}(f_{\theta,\phi}(\mathbf{z}_{t_k}, \mathbf{c})))$
11:     ## Compute Boundary loss
12:     $\mathcal{L}_{bound} \leftarrow d(f_{\theta,\phi}(\mathbf{z}, \mathbf{c}), \mathbf{x})$
13:     ## Compute Total Loss and Update
14:     $\mathcal{L}_{total} \leftarrow \mathcal{L}_{con} + \lambda\mathcal{L}_{bound}$
15:     Update $\phi$ using $\nabla_\phi \mathcal{L}_{total}$
16:     # Dual Step:
17:     Update $\lambda$ according to Eq. (12).
18: **end for**

---

**Definition 1** (Noise Consistency Training). *Given a fixed margin $\xi$, the general optimization can be transformed into the following:*

$$
\begin{aligned}
\min_\phi \quad & \mathbb{E}_{\mathbf{z},\mathbf{c}|\mathbf{z},\epsilon} L_{\mathrm{con}}(\phi) = d(f_{\theta,\phi}(\mathbf{z}_{t_n}, \mathbf{c}), \mathrm{sg}(f_{\theta,\phi}(\mathbf{z}_{t_{n-1}}, \mathbf{c}))) \\
\mathrm{s.t.} \quad & L_{\mathrm{bound}}(\phi) = \mathbb{E}_{\mathbf{z},\mathbf{c}|\mathbf{z},\epsilon} d(f_{\theta,\phi}(\mathbf{z}, \mathbf{c}), f_\theta(\mathbf{z})) < \xi,
\end{aligned}
\tag{9}
$$

*where $\mathbf{z}, \epsilon \sim \mathcal{N}(0, I)$, $\mathbf{c} \sim p(\mathbf{c}|f_\theta(\mathbf{z}))$, $\mathbf{z}_{t_n} = \sqrt{1 - \sigma_{t_n}^2}\mathbf{z} + \sigma_{t_n}\epsilon$ and $\mathbf{z}_{t_{n+1}} = \sqrt{1 - \sigma_{t_{n+1}}^2}\mathbf{z} + \sigma_{t_{n+1}}\epsilon$.*

The constrained optimization problem presented in Definition 1 is hard to optimize directly. We therefore reformulate it as a corresponding saddle-point problem:

$$
\max_\lambda \min_\phi \left\{ L_{\mathrm{con}}(\phi) + \lambda L_{\mathrm{bound}}(\phi) \right\}, \quad \lambda \geq 0.
\tag{10}
$$

**Concrete Algorithm** To efficiently optimize this saddle-point problem, we employ the primal-dual algorithm tailored for the saddle-point problem, which alternates between updating the primal variables $\phi$ and the dual variable $\lambda$. Specifically, in the *primal* step, for a given dual variable $\lambda$, the algorithm minimizes the corresponding empirical Lagrangian with respect to $\phi$ under a given dual variable $\lambda$, i.e.,

$$
\phi_{k+1} := \arg\min_\phi \left\{ L_{\mathrm{con}}(\phi_k) + \lambda L_{\mathrm{bound}}(\phi_k) \right\}
\tag{11}
$$

In practice, this update for $\phi$ is performed using stochastic gradient descent. Subsequently, in the *dual* step, we update the dual variable $\lambda$ as follows:

$$
\lambda_{t+1} := \max \left\{ \lambda_t + \eta \cdot (L_{\mathrm{con}} - \xi), 0 \right\},
\tag{12}
$$

where $\eta$ is the learning rate for the dual update.

Algorithm 1 provides the pseudo-code for our primal-dual optimization of the adapter parameters $\phi$. In contrast to the direct application of stochastic gradient descent in Eq. (10), the primal-dual algorithm dynamically adjusts $\lambda$. This avoids an extra hyper-parameter tuning and can provide an early-stopping condition (e.g., $\lambda = 0$). Additionally, convergence is guaranteed under sufficiently long training and an adequately small step size [34].

Table 1: Comparison of machine metrics of different methods for Canny, HED, Depth and 8× Super Resolution tasks. The mark † denotes our reimplementation with the same one-step generator as used in NCT.

| Method | NFE↓ | Canny FID↓ | Canny Consistency↓ | HED FID↓ | HED Consistency↓ | Depth FID↓ | Depth Consistency↓ | 8× Super Resolution FID↓ | 8× Super Resolution Consistency↓ | Avg FID↓ | Avg Consistency↓ |
|---|---|---|---|---|---|---|---|---|---|---|---|
| ControlNet | 50 | 14.48 | 0.113 | 19.21 | 0.101 | **15.25** | 0.093 | **11.93** | 0.065 | 15.22 | 0.093 |
| DI + ControlNet | 1 | 22.74 | 0.141 | 28.04 | 0.113 | 22.49 | 0.097 | 15.57 | 0.126 | 22.21 | 0.119 |
| JDM† | 1 | 14.35 | 0.122 | 16.75 | **0.055** | 16.71 | 0.093 | 13.23 | 0.068 | 15.26 | 0.085 |
| **NCT (Ours)** | 1 | **13.67** | **0.110** | **14.96** | 0.060 | 16.45 | **0.088** | 12.17 | **0.053** | **14.31** | **0.078** |

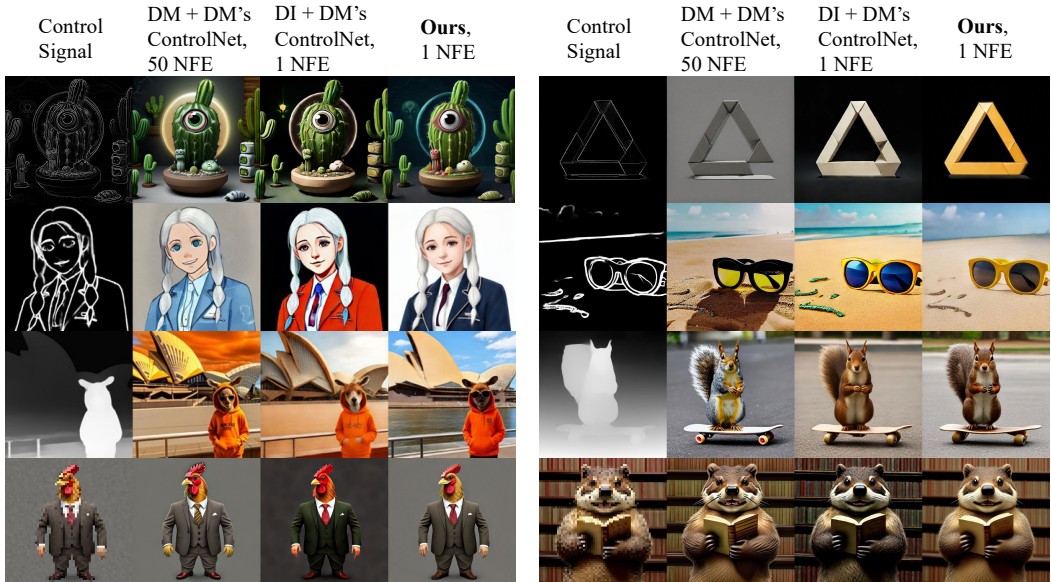

Figure 2: Qualitative comparisons on controllable generation across different control signals against competing methods.

# 4 Experiments

## 4.1 Controllable Generation

**Experimental Setup.** All models are trained on an internally collected dataset. The one-step generator was initialized using weights from Stable Diffusion 1.5 [35]. Subsequently, the one-step generator was pre-trained using the Diff-Instruct [3]. The ControlNet was initialized following the procedure outlined in its original publication [15]. An Exponential Moving Average (EMA) with a decay rate of 0.9999 was applied to the ControlNet parameters, denoted as $\phi$.

To evaluate the performance of our proposed method in one-step controllable generation, we employed four distinct conditioning signals: Canny edges [36], HED (Holistically-Nested Edge Detection) boundaries [37], depth maps, and lower-resolution images.

**Evaluation Metric.** Image quality was assessed using the Fréchet Inception Distance (FID) [38]. Specifically, the FID score was computed by comparing images generated by the base diffusion model without controls against images generated with the incorporation of the aforementioned conditional inputs. The consistency metric for measuring controllability is quantified between the conditioning input c and the condition extracted from the generated image $h(\mathbf{x})$, as formulated below:

$$\text{Consistency} = ||h(\mathbf{x}) - \mathbf{c}||_1, \tag{13}$$

where $h(\cdot)$ represents the function used to extract the conditioning information (e.g., Canny edge detector, depth estimator) from a generated image $\mathbf{x}$, and c is the target conditional input. Furthermore, to assess computational efficiency, we report the NFE required to generate a single image.

**Quantitative Results.** We conduct comprehensive evaluations, benchmarking our proposed approach against three established baseline methods: (1) the standard diffusion model with ControlNet; (2) a

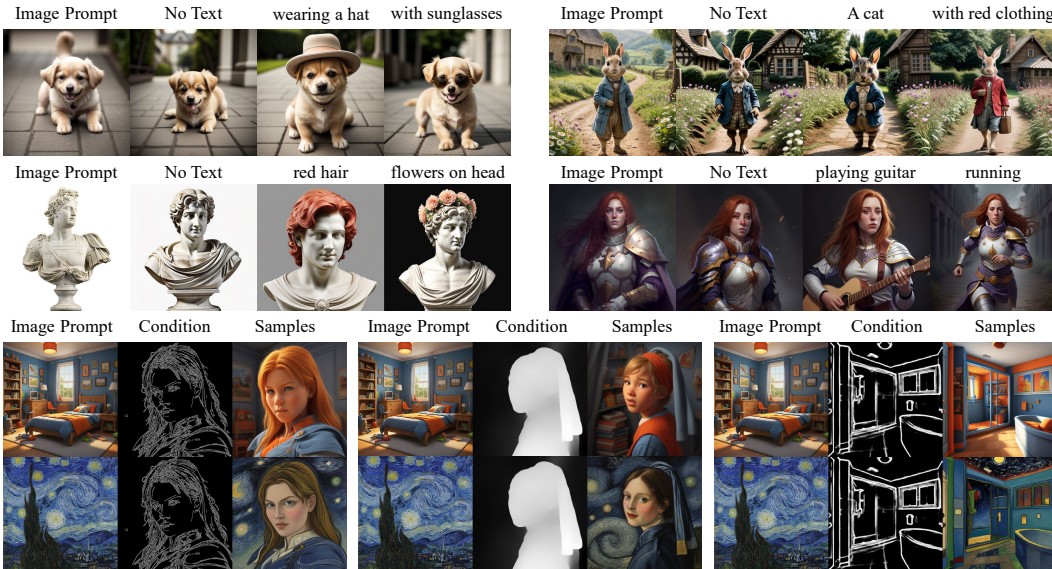

Figure 3: Visual samples of image-reference generations. The samples are generated by our NCT with 1NFE.

pre-trained one-step generator integrated with the DM's ControlNet; and (3) a crafted ControlNet specifically trained for a one-step generator trained via JDM distillation [9]. *Notably, the JDM approach necessitates an additional, computationally intensive distillation phase to incorporate control mechanisms. This step is redundant given that the one-step generator has already undergone a distillation process. In contrast, our method is tailored for one-step generators, obviating the need for further distillation and thereby enhancing computational efficiency.* The quantitative results, presented in Table 1, assess both image fidelity (FID) and adherence to conditional inputs across diverse control tasks. Our proposed method achieves a remarkable reduction in the number of function evaluations (NFEs) from 50 to 1, while concurrently maintaining or surpassing the performance metrics of the baselines. Specifically, our approach demonstrates superior FID scores and stronger consistency measures across various conditioning tasks, signifying enhanced image quality and more precise alignment with control conditions. These findings collectively establish that our method achieves a superior trade-off between computational efficiency and sample quality in controlled image generation. It delivers state-of-the-art performance with substantially reduced computational overhead and a more streamlined training pipeline.

**Qualitative Comparison.** A qualitative comparison of our method against baselines is presented in Fig. 2, comparing standard ControlNet and DI+ControlNet which does not require additional distillation. Visual results reveal that while the standard DM's ControlNet can impart high-level control to one-step generators, this integration frequently results in a discernible degradation of image quality. In stark contrast, our approach, which involves customized training for adding new controls to one-step generator, consistently produces images of significantly higher fidelity. These visual results substantiate the efficacy of our proposed methodology, suggesting its capability to implicitly learn the conditional distribution $p(\mathbf{x}|\mathbf{c})$ through our novel noise consistency training.

## 4.2 Image Prompted Generation

**Experiment Setting.** The pre-trained one-step generator remains consistent with that employed in the prior experiments. We employ the IP-Adapter [39] architecture to serve as the adapter for injecting image prompts. Following IP-Adapter, we use OpenCLIP ViT-H/14 as the image encoder..

**Quantitative Comparison.** Our method is quantitatively benchmarked against the original IP-Adapter. Following IP-Adapter [39], we generate four images conditioned on each image prompt, for every sample in the COCO-2017-5k dataset [40]. Alignment with the image condition is assessed using two established metrics: 1) CLIP-I: The cosine similarity between the CLIP image embeddings of the generated images and the respective image prompt; 2) CLIP-T: The CLIP Score measuring

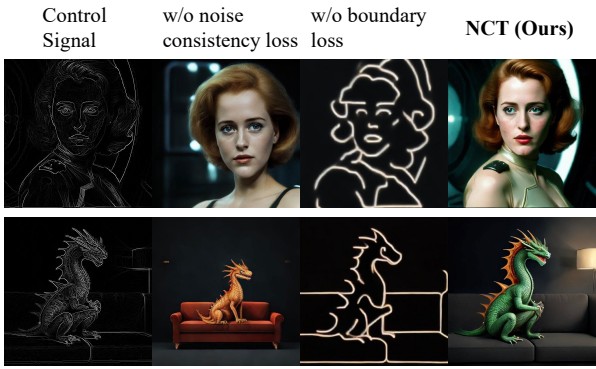

| | Control Signal | w/o noise consistency loss | w/o boundary loss | NCT (Ours) |

Figure 4: Both boundary loss and noise consistency loss are crucial to our NCT. Without Boundary loss, the model's distribution collapses. Without noise consistency loss, the model ignores the injected condition.

| Method | NFE↓ | Clip-T↑ | Clip-I↑ |
|---|---|---|---|
| IP-Adapter[†] | 100 | 0.588 | 0.828 |
| JDM | 1 | 0.585 | 0.826 |
| Ours | 1 | 0.593 | 0.821 |

Table 2: Comparison of machine metrics of different methods regarding image-prompted generation. The mark [†] denotes that the result is taken from the official report.

| Method | FID↓ | Con.↓ |
|---|---|---|
| Ours | **13.67** | **0.110** |
| w/o noise consistency loss | 20.56 | 0.165 |
| w/o boundary loss | 216.93 | 0.113 |
| w/o primal-dual | 14.13 | 0.117 |

Table 3: Ablation study on proposed components in our NCT.

the similarity between the generated images and the captions corresponding to the image prompts. The quantitative results, summarized in Table 2, reveal that our Noise Consistency Training (NCT) method achieves performance comparable to the original IP-Adapter (which necessitates 100 NFEs) on both CLIP-I and CLIP-T metrics. Crucially, NCT attains this level of performance with only a single NFE, signifying an approximate 100-fold improvement in computational efficiency.

**Multi-modal Prompts.** Our investigations indicate that NCT can concurrently process both image and textual prompts. Fig. 3 illustrates generation outcomes achieved through the use of such multimodal inputs. As demonstrated, the integration of supplementary text prompts facilitates the generation of more diverse visual outputs. This allows for capabilities such as attribute modification and scene alteration based on textual descriptions, relative to the content of the primary image prompt.

**Structure Control.** We observe that NCT permits the test-time compatibility of adapters designed for image prompting with those designed for controllable generation. This enables the generation of images based on image prompts while jointly incorporating additional structural or conditional controls, as shown in Fig. 3. Such test-time compatibility underscores the inherent flexibility and potential of NCT for training distinct adapters for a one-step generator, which can subsequently be combined effectively during the inference stage.

### 4.3 Ablation Study

**The Effect of Noise Consistency Loss.** The noise consistency loss is crucial to force adapter $\phi$ to learn condition c. Without the loss, it can be seen that the consistency metric degrades severely, and the generated samples do not follow the condition at all. This is because the adapter $\phi$ is trained on fully-coupled $(\mathbf{z}, \mathbf{c})$ pairs, allowing it find find a shortcut solution that directly ignores the learnable parameters to satisfy the boundary loss.

**The Effect of Boundary Loss.** The boundary loss can constrain the output of the generator $f_{\theta,\phi}$ in the image domain. Without the loss, although the generator can still learns some conditions, its generated samples entirely collapse as indicated by the FID and visual samples.

**The Effect of Primal-Dual.** We use primal-dual since it is crafted for solving the constrained problem, while it owns theoretical guarantees and dynamically balances the noise consistency loss and boundary loss. We empirically validate its effectiveness, it can be seen that without primal-dual, the performance slightly degrades regarding both fidelity and condition alignment.

## 5 Conclusion

This paper addressed the critical challenge of efficiently incorporating new controls into pre-trained one-step generative models, a key bottleneck in the rapidly evolving field of AIGC. We introduced Noise Consistency Training (NCT), a novel and lightweight approach that empowers existing one-step

generators with new conditioning capabilities without the need for retraining the base diffusion model or additional diffusion distillation or accessing the original training dataset. By operating in the noise space and leveraging a carefully formulated noise-space consistency loss, NCT effectively aligns the adapted generator with the desired control signals. Our proposed NCT framework offers significant advantages in terms of modularity, data efficiency, and ease of deployment. The experimental results across diverse control scenarios robustly demonstrate that NCT achieves state-of-the-art performance in controllable, single-step generation. It surpasses existing multi-step and distillation-based methods in both the quality of the generated content and computational efficiency.

## Acknowledgments

Jing Tang's work is partially supported by National Key R&D Program of China under Grant No. 2024YFA1012700 and No. 2023YFF0725100, by the National Natural Science Foundation of China (NSFC) under Grant No. 62402410 and No. U22B2060, by Guangdong Provincial Project (No. 2023QN10X025), by Guangdong Basic and Applied Basic Research Foundation under Grant No. 2023A1515110131, by Guangzhou Municipal Science and Technology Bureau under Grant No. 2024A04J4454, by Guangzhou Municipal Education Bureau (No. 2024312263), and by Guangzhou Industrial Information and Intelligent Key Laboratory Project (No. 2024A03J0628) and Guangzhou Municipal Key Laboratory of Financial Technology Cutting-Edge Research (No. 2024A03J0630).

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

# A  Theoretical Foundation of Noise Consistency Training

This section establishes the theoretical foundation: it begins with definitions and the mathematical setup (A.1), then introduces key lemmas (A.2) that collectively build the necessary mathematical foundation—by defining critical distribution relationships and an input interpolation path—for formulating the conditions and proving the main theorem (A.3). Specifically, Lemma 1, Lemma 2, and Theorem 1 presented in the main paper are proved in Lemma A.3, Remark 1, and Theorem A.1 respectively in this section.

## A.1  Definition and Setup

- **Latent Distribution:** The latent distribution $\Gamma$ is the standard Gaussian measure on $(\mathbb{R}^m, \mathcal{B}(\mathbb{R}^m))$. The measure $\Gamma$ has density $\gamma(\mathbf{z})$ w.r.t. $\mathrm{d}\mathbf{z}$, so $\mathrm{d}\Gamma(\mathbf{z}) = \gamma(\mathbf{z})\mathrm{d}\mathbf{z}$. $\Gamma$ is a probability measure: $\int_{\mathbb{R}^m} \gamma(\mathbf{z})\mathrm{d}\mathbf{z} = 1$.

- **Implicit Generator:** $f_\theta : \mathbb{R}^m \to \mathbb{R}^n$ is a measurable function.

- **Data Distribution:** $P_\theta$ on $(\mathbb{R}^n, \mathcal{B}(\mathbb{R}^n))$ is the push-forward $P_\theta = f_{\theta\#}\Gamma$. It has density $p(\mathbf{x})$ w.r.t. $\mathrm{d}\mathbf{x}$, so $\mathrm{d}P_\theta(\mathbf{x}) = p_\theta(\mathbf{x})\mathrm{d}\mathbf{x}$. Since $\Gamma$ is a probability measure, $P_\theta$ is also a probability measure: $\int_{\mathbb{R}^n} p_\theta(\mathbf{x})\mathrm{d}\mathbf{x} = 1$.

- **Condition:.** Let $(\mathcal{C}, \mathcal{B}_\mathcal{C}, \mu_\mathcal{C})$ be a measure space for the conditions. $\mathcal{B}_\mathcal{C}$ is a $\sigma$-algebra on $\mathcal{C}$, and $\mu_\mathcal{C}$ is a reference measure (e.g., Lebesgue measure if $\mathcal{C} = \mathbb{R}^k$ (such as Canny Edge), or counting measure if $\mathcal{C}$ is discrete) (such as class labels). For each $\mathbf{x} \in \mathbb{R}^n$, $p(\cdot|\mathbf{x})$ is a probability measure on $(\mathcal{C}, \mathcal{B}_\mathcal{C})$. We assume it has a density $p(\mathbf{c}|\mathbf{x})$ with respect to $\mu_\mathcal{C}$. Thus, for any $\mathbf{x} \in \mathbb{R}^n$: $\int_\mathcal{C} p(\mathbf{c}|\mathbf{x})\mathrm{d}\mu_\mathcal{C}(\mathbf{c}) = 1$.

- **Combined Map:** $T = f_\theta \times \mathrm{id} : \mathbb{R}^m \times \mathcal{C} \to \mathbb{R}^n \times \mathcal{C}$, $T(\mathbf{z}, \mathbf{c}) = (f_\theta(\mathbf{z}), \mathbf{c})$. Since $f_\theta$ and $\mathrm{id}$ are measurable, $T$ is measurable with respect to the product $\sigma$-algebras $\mathcal{B}(\mathbb{R}^m) \otimes \mathcal{B}_\mathcal{C}$ and $\mathcal{B}(\mathbb{R}^n) \otimes \mathcal{B}_\mathcal{C}$.

- **Implicit Generator with Condition:** $f_{\theta,\phi} : \mathbb{R}^m \times \mathcal{C} \to \mathbb{R}^n$ be a measurable function.

- **Combined Map with Condition:** We define a new map $T_\phi : \mathbb{R}^m \times \mathcal{C} \to \mathbb{R}^n \times \mathcal{C}$ as $T_\phi(\mathbf{z}, \mathbf{c}) = (f_{\theta,\phi}(\mathbf{z}, \mathbf{c}), \mathbf{c})$.

- **Marginal Condition Density:** We define the marginal probability density $p(\mathbf{c})$ of the condition c as:

$$p(\mathbf{c}) = \int_{\mathbb{R}^n} p_\theta(\mathbf{x})p(\mathbf{c}|\mathbf{x})\mathrm{d}\mathbf{x} = \int_{\mathbb{R}^m} \gamma(\mathbf{z})p(\mathbf{c}|f_\theta(\mathbf{z}))\mathrm{d}\mathbf{z}$$

  This is a probability density with respect to $\mu_\mathcal{C}$, i.e., $\int_\mathcal{C} p(\mathbf{c})\mathrm{d}\mu_\mathcal{C}(\mathbf{c}) = 1$.

- **Initial Coupled Latent-Condition Distribution**: Density $\nu(\mathbf{z}, \mathbf{c}) = \gamma(\mathbf{z})p(\mathbf{c}|f_\theta(\mathbf{z}))$ w.r.t. $\mathrm{d}\mathbf{z}\mathrm{d}\mu_\mathcal{C}(\mathbf{c})$.

- **Independent Latent-Condition Coupling:** We define the probability measure $\rho$ on the input space $(\mathbb{R}^m \times \mathcal{C}, \mathcal{B}(\mathbb{R}^m) \otimes \mathcal{B}_\mathcal{C})$ by its density with respect to the reference measure $\mathrm{d}\mathbf{z}\,\mathrm{d}\mu_\mathcal{C}(\mathbf{c})$:
$$\mathrm{d}\rho(\mathbf{z}, \mathbf{c}) = \gamma(\mathbf{z})p(\mathbf{c})\mathrm{d}\mathbf{z}\,\mathrm{d}\mu_\mathcal{C}(\mathbf{c})$$
  Here, $\gamma(\mathbf{z})$ is the density of the standard Gaussian measure $\Gamma$ on $\mathbb{R}^m$. The measure $\rho$ corresponds to sampling $\mathbf{z} \sim \Gamma$ independently from sampling $\mathbf{c} \sim p(\mathbf{c})$.

- **Target Data-Condition Distribution** $\eta$: Density $p_\eta(\mathbf{x}, \mathbf{c}) = p_\theta(\mathbf{x})p(\mathbf{c}|\mathbf{x})$ w.r.t. $\mathrm{d}\mathbf{x}\mathrm{d}\mu_\mathcal{C}(\mathbf{c})$.

- **MMD (Maximum Mean Discrepancy)**: $\mathrm{MMD}^2(P, Q)$ is a metric between probability distributions $P$ and $Q$. For a characteristic kernel, $\mathrm{MMD}^2(P, Q) = 0 \iff P = Q$.

## A.2  Lemmas

**Lemma A.1.** *Let $\Gamma$ be the standard Gaussian measure on $\mathbb{R}^m$ with density $\gamma(\mathbf{z})$ with respect to the Lebesgue measure $\mathrm{d}\mathbf{z}$. Let $f_\theta : \mathbb{R}^m \to \mathbb{R}^n$ be a measurable function, and let $P_\theta = f_{\theta\#}\Gamma$ be the push-forward measure on $\mathbb{R}^n$, assumed to have a density $p_\theta(\mathbf{x})$ with respect to the Lebesgue measure $\mathrm{d}\mathbf{x}$. Let $(\mathcal{C}, \mathcal{B}_\mathcal{C}, \mu_\mathcal{C})$ be a measure space for conditions, and let $p(\mathbf{c}|\mathbf{x})$ be a conditional probability density on $\mathcal{C}$ with respect to $\mu_\mathcal{C}$ for each $\mathbf{x} \in \mathbb{R}^n$, such that $\int_\mathcal{C} p(\mathbf{c}|\mathbf{x})\mathrm{d}\mu_\mathcal{C}(\mathbf{c}) = 1$.*

*Define the measure $\nu$ on $\mathbb{R}^m \times \mathcal{C}$ by its density with respect to $\mathrm{d}\mathbf{z}\mathrm{d}\mu_\mathcal{C}(\mathbf{c})$:*

$$\mathrm{d}\nu(\mathbf{z}, \mathbf{c}) = \gamma(\mathbf{z})p(\mathbf{c}|f_\theta(\mathbf{z}))\mathrm{d}\mathbf{z}\mathrm{d}\mu_\mathcal{C}(\mathbf{c})$$

*Define the map $T = f_\theta \times id : \mathbb{R}^m \times \mathcal{C} \to \mathbb{R}^n \times \mathcal{C}$ by $T(\mathbf{z}, \mathbf{c}) = (f_\theta(\mathbf{z}), \mathbf{c})$.*

*Then the push-forward measure $T_{\#}\nu$ on $\mathbb{R}^n \times \mathcal{C}$ has the density $p_\theta(\mathbf{x})p(\mathbf{c}|\mathbf{x})$ with respect to $\mathrm{d}\mathbf{x}\mathrm{d}\mu_\mathcal{C}(\mathbf{c})$. That is,*

$$(f_\theta \times id)_{\#}(\gamma(\mathbf{z})p(\mathbf{c}|f_\theta(\mathbf{z}))\mathrm{d}\mathbf{z}\mathrm{d}\mu_\mathcal{C}(\mathbf{c})) = p_\theta(\mathbf{x})p(\mathbf{c}|\mathbf{x})\mathrm{d}\mathbf{x}\mathrm{d}\mu_\mathcal{C}(\mathbf{c})$$

*Proof.* We want to show $T_{\#}\nu = \eta$. By definition of equality of measures, it suffices to show that for any bounded, measurable test function $\Phi : \mathbb{R}^n \times \mathcal{C} \to \mathbb{R}$:

$$\int_{\mathbb{R}^n \times \mathcal{C}} \Phi(\mathbf{x}, \mathbf{c})\mathrm{d}(T_{\#}\nu)(\mathbf{x}, \mathbf{c}) = \int_{\mathbb{R}^n \times \mathcal{C}} \Phi(\mathbf{x}, \mathbf{c})\mathrm{d}\eta(\mathbf{x}, \mathbf{c})$$

We start with the left-hand side (LHS). Using the change of variables formula for push-forward measures:

$$\begin{aligned} LHS &= \int_{\mathbb{R}^n \times \mathcal{C}} \Phi(\mathbf{x}, \mathbf{c})\mathrm{d}(T_{\#}\nu)(\mathbf{x}, \mathbf{c}) \\ &= \int_{\mathbb{R}^m \times \mathcal{C}} \Phi(T(\mathbf{z}, \mathbf{c}))\mathrm{d}\nu(\mathbf{z}, \mathbf{c}) \quad \text{(Change of Variables)} \\ &= \int_{\mathbb{R}^m \times \mathcal{C}} \Phi(f_\theta(\mathbf{z}), \mathbf{c})\gamma(\mathbf{z})p(\mathbf{c}|f_\theta(\mathbf{z}))\mathrm{d}\mathbf{z}\mathrm{d}\mu_\mathcal{C}(\mathbf{c}) \quad \text{(Substitute } T \text{ and density of } \nu\text{)} \\ &= \int_{\mathbb{R}^m} \gamma(\mathbf{z}) \left[\int_\mathcal{C} \Phi(f_\theta(\mathbf{z}), \mathbf{c})p(\mathbf{c}|f_\theta(\mathbf{z}))\mathrm{d}\mu_\mathcal{C}(\mathbf{c})\right] \mathrm{d}\mathbf{z} \quad \text{(Fubini's Theorem)} \end{aligned}$$

The application of Fubini's theorem is justified because $\Phi$ is bounded, $\gamma(\mathbf{z}) \geq 0$, $p(\mathbf{c}|f_\theta(\mathbf{z})) \geq 0$, and $\nu$ is a finite (probability) measure.

Let's define an auxiliary function $g : \mathbb{R}^n \to \mathbb{R}$ as:

$$g(\mathbf{y}) = \int_\mathcal{C} \Phi(\mathbf{y}, \mathbf{c})p(\mathbf{c}|\mathbf{y})\mathrm{d}\mu_\mathcal{C}(\mathbf{c})$$

Since $\Phi$ is bounded (say $|\Phi| \leq M$) and $\int_\mathcal{C} p(\mathbf{c}|\mathbf{y})\mathrm{d}\mu_\mathcal{C}(\mathbf{c}) = 1$, $g(\mathbf{y})$ is also bounded ($|g(\mathbf{y})| \leq M$). If $\Phi$ is $\mathcal{B}(\mathbb{R}^n) \otimes \mathcal{B}_\mathcal{C}$-measurable and $p(\mathbf{c}|\mathbf{y})$ defines a measurable transition kernel, then $g$ is $\mathcal{B}(\mathbb{R}^n)$-measurable.

Substituting $g$ into our integral expression:

$$LHS = \int_{\mathbb{R}^m} g(f_\theta(\mathbf{z}))\gamma(\mathbf{z})\mathrm{d}\mathbf{z}$$

Now, recall the definition of the push-forward measure $P_\theta = f_{\#}\Gamma$. For any bounded, measurable function $h : \mathbb{R}^n \to \mathbb{R}$:

$$\int_{\mathbb{R}^n} h(\mathbf{x})\mathrm{d}P_\theta(\mathbf{x}) = \int_{\mathbb{R}^m} h(f_\theta(\mathbf{z}))\mathrm{d}\Gamma(\mathbf{z})$$

In terms of densities:

$$\int_{\mathbb{R}^n} h(\mathbf{x})p_\theta(\mathbf{x})\mathrm{d}\mathbf{x} = \int_{\mathbb{R}^m} h(f_\theta(\mathbf{z}))\gamma(\mathbf{z})\mathrm{d}\mathbf{z}$$

Applying this identity with $h = g$:

$$\int_{\mathbb{R}^m} g(f_\theta(\mathbf{z}))\gamma(\mathbf{z})\mathrm{d}\mathbf{z} = \int_{\mathbb{R}^n} g(\mathbf{x})p_\theta(\mathbf{x})\mathrm{d}\mathbf{x}$$

So,

$$LHS = \int_{\mathbb{R}^n} g(\mathbf{x})p_\theta(\mathbf{x})\mathrm{d}\mathbf{x}$$

Now, substitute back the definition of $g(\mathbf{x})$:

$$LHS = \int_{\mathbb{R}^n} \left[\int_\mathcal{C} \Phi(\mathbf{x}, \mathbf{c})p(\mathbf{c}|\mathbf{x})\mathrm{d}\mu_\mathcal{C}(\mathbf{c})\right] p_\theta(\mathbf{x})\mathrm{d}\mathbf{x}$$

Applying Fubini's Theorem again (justified as before):

$$LHS = \int_{\mathbb{R}^n \times \mathcal{C}} \Phi(\mathbf{x}, \mathbf{c}) p_\theta(\mathbf{x}) p(\mathbf{c}|\mathbf{x}) \mathrm{d}\mathbf{x} \mathrm{d}\mu_\mathcal{C}(\mathbf{c})$$

This is precisely the integral with respect to the target measure $\eta$:

$$LHS = \int_{\mathbb{R}^n \times \mathcal{C}} \Phi(\mathbf{x}, \mathbf{c}) \mathrm{d}\eta(\mathbf{x}, \mathbf{c})$$

Since we have shown that $\int \Phi \mathrm{d}(T_\# \nu) = \int \Phi \mathrm{d}\eta$ for all bounded, measurable test functions $\Phi$, the measures must be equal:

$$T_\# \nu = \eta$$

$\square$

**Lemma A.2** (Boundary Loss). *Let $f_{\theta,\phi} : \mathbb{R}^m \times \mathcal{C} \to \mathbb{R}^n$ be a measurable function. Let $\nu$ be the measure on $\mathbb{R}^m \times \mathcal{C}$ with density $\gamma(\mathbf{z}) p(\mathbf{c}|f_\theta(\mathbf{z}))$ w.r.t. $\mathrm{d}\mathbf{z}\mathrm{d}\mu_\mathcal{C}(\mathbf{c})$. Let $d$ be a distance metric on $\mathbb{R}^n$. If the boundary loss*

$$\mathbb{E}_{(\mathbf{z},\mathbf{c}) \sim \nu}[d(f_{\theta,\phi}(\mathbf{z}, \mathbf{c}), f_\theta(\mathbf{z}))] = 0$$

*then:*

*The push-forward measure $\eta_\phi = (T_\phi)_\# \nu$ is equal to the target measure $\eta = T_\# \nu$. This means the joint distribution $p_{\theta,\phi}(\mathbf{x}, \mathbf{c})$ induced by $f_{\theta,\phi}$ is $p_\theta(\mathbf{x}) p(\mathbf{c}|\mathbf{x})$, i.e., $\eta_\phi = \eta$.*

*Proof.* The condition is $\mathbb{E}_{(\mathbf{z},\mathbf{c}) \sim \nu}[d(f_{\theta,\phi}(\mathbf{z}, \mathbf{c}), f_\theta(\mathbf{z}))] = 0$. Since $d(a, b) \geq 0$ for any $a, b \in \mathbb{R}^n$, and $d(a, b) = 0$ if and only if $a = b$, the expectation of this non-negative quantity being zero implies that the integrand must be zero $\nu$-almost everywhere. That is,

$$d(f_{\theta,\phi}(\mathbf{z}, \mathbf{c}), f_\theta(\mathbf{z})) = 0 \quad \text{for } \nu\text{-a.e. } (\mathbf{z}, \mathbf{c})$$

This implies

$$f_{\theta,\phi}(\mathbf{z}, \mathbf{c}) = f_\theta(\mathbf{z}) \quad \text{for } \nu\text{-a.e. } (\mathbf{z}, \mathbf{c})$$

We want to show that $\eta_\phi = (T_\phi)_\# \nu$ is equal to $\eta = T_\# \nu$. Recall the definitions of the maps:

$$T(\mathbf{z}, \mathbf{c}) = (f_\theta(\mathbf{z}), \mathbf{c})$$

$$T_\phi(\mathbf{z}, \mathbf{c}) = (f_{\theta,\phi}(\mathbf{z}, \mathbf{c}), \mathbf{c})$$

Since $f_{\theta,\phi}(\mathbf{z}, \mathbf{c}) = f_\theta(\mathbf{z})$ for $\nu$-a.e. $(\mathbf{z}, \mathbf{c})$, it follows directly that the maps $T_\phi$ and $T$ are equal $\nu$-almost everywhere:

$$T_\phi(\mathbf{z}, \mathbf{c}) = (f_{\theta,\phi}(\mathbf{z}, \mathbf{c}), \mathbf{c}) = (f_\theta(\mathbf{z}), \mathbf{c}) = T(\mathbf{z}, \mathbf{c}) \quad \text{for } \nu\text{-a.e. } (\mathbf{z}, \mathbf{c})$$

If two measurable maps $T$ and $T_\phi$ are equal $\nu$-a.e., their push-forward measures $T_\# \nu$ and $(T_\phi)_\# \nu$ are identical. Let $\Psi : \mathbb{R}^n \times \mathcal{C} \to \mathbb{R}$ be any bounded, measurable test function.

$$\int_{\mathbb{R}^n \times \mathcal{C}} \Psi(\mathbf{x}, \mathbf{c}) \mathrm{d}((T_\phi)_\# \nu)(\mathbf{x}, \mathbf{c}) = \int_{\mathbb{R}^m \times \mathcal{C}} \Psi(T_\phi(\mathbf{z}, \mathbf{c})) \mathrm{d}\nu(\mathbf{z}, \mathbf{c})$$

$$= \int_{\mathbb{R}^m \times \mathcal{C}} \Psi(T(\mathbf{z}, \mathbf{c})) \mathrm{d}\nu(\mathbf{z}, \mathbf{c}) \quad (\text{since } T_\phi = T \ \nu\text{-a.e. and } \Psi \text{ is bounded})$$

$$= \int_{\mathbb{R}^n \times \mathcal{C}} \Psi(\mathbf{x}, \mathbf{c}) \mathrm{d}(T_\# \nu)(\mathbf{x}, \mathbf{c})$$

Since this holds for all bounded measurable $\Psi$, we have $(T_\phi)_\# \nu = T_\# \nu$. From Lemma 1, we know $T_\# \nu = \eta$, where $\eta$ has density $p_\theta(\mathbf{x}) p(\mathbf{c}|\mathbf{x})$ with respect to $\mathrm{d}\mathbf{x}\mathrm{d}\mu_\mathcal{C}(\mathbf{c})$. Therefore, $\eta_\phi = (T_\phi)_\# \nu = \eta$. $\square$

**Lemma A.3** (Interpolation of Joint Latent-Condition Distributions (Lemma 1 in main paper)). *Let $\gamma(\mathbf{z})$ be the density of the standard Gaussian measure on $\mathbb{R}^m$. Let $f_\theta : \mathbb{R}^m \to \mathbb{R}^n$ be a measurable function. Let $p(\mathbf{c}|\mathbf{x})$ be a conditional probability density on $\mathcal{C}$ (with respect to a reference measure $\mu_\mathcal{C}$) for each $\mathbf{x} \in \mathbb{R}^n$. Define the marginal condition density $p(\mathbf{c})$ as:*

$$p(\mathbf{c}) = \int_{\mathbb{R}^m} \gamma(\mathbf{z}') p(\mathbf{c}|f_\theta(\mathbf{z}')) \mathrm{d}\mathbf{z}'$$

*Assume $p(c) > 0$ for $\mu_C$-almost every $c$ in the support of interest. Define the conditional latent density $p_{data}(\mathbf{z}_0|c)$ as:*

$$p_{data}(\mathbf{z}_0|c) = \frac{\gamma(\mathbf{z}_0)p(c|f_\theta(\mathbf{z}_0))}{p(c)}$$

*Consider a time-dependent process for $t \in [0, 1]$ where $\mathbf{z}_t$ is generated from $\mathbf{z}_0 \sim p_{data}(\cdot|c)$ by:*

$$\mathbf{z}_t = \alpha_t\mathbf{z}_0 + \sigma_t\boldsymbol{\epsilon}, \quad \text{where } \boldsymbol{\epsilon} \sim \mathcal{N}(0, I_m) \text{ independent of } \mathbf{z}_0 \text{ and } c.$$

*The coefficients $\alpha_t, \sigma_t \in \mathbb{R}$ satisfy:*

- *$\alpha_0 = 1, \sigma_0 = 0$*

- *$\alpha_1 = 0, \sigma_1 = 1$*

- *$\alpha_t$ is monotonically decreasing, $\sigma_t$ is monotonically increasing.*

*Let $q_t(\mathbf{z}_t|\mathbf{z}_0) = \mathcal{N}(\mathbf{z}_t; \alpha_t\mathbf{z}_0, \sigma_t^2 I_m)$ be the density of $\mathbf{z}_t$ given $\mathbf{z}_0$. Define the conditional density $p_t(\mathbf{z}|c)$ as:*

$$p_t(\mathbf{z}|c) = \int_{\mathbb{R}^m} q_t(\mathbf{z}|\mathbf{z}_0)p_{data}(\mathbf{z}_0|c)\mathrm{d}\mathbf{z}_0$$

*And the joint density $p_t(\mathbf{z}, c)$ on $\mathbb{R}^m \times C$ (with respect to $\mathrm{d}\mathbf{z}\mathrm{d}\mu_C(c)$) as:*

$$p_t(\mathbf{z}, c) = p_t(\mathbf{z}|c)p(c)$$

*Then,*

1. *At $t = 0$, the joint density is $p_0(\mathbf{z}, c) = \gamma(\mathbf{z})p(c|f_\theta(\mathbf{z}))$.*

2. *At $t = 1$, the joint density is $p_1(\mathbf{z}, c) = \gamma(\mathbf{z})p(c)$.*

*Proof.* The joint density at time $t$ is given by $p_t(\mathbf{z}, c) = p_t(\mathbf{z}|c)p(c)$. Substituting the definition of $p_t(\mathbf{z}|c)$:

$$p_t(\mathbf{z}, c) = p(c)\int_{\mathbb{R}^m} \mathcal{N}(\mathbf{z}; \alpha_t\mathbf{z}_0, \sigma_t^2 I_m)p_{\text{data}}(\mathbf{z}_0|c)\mathrm{d}\mathbf{z}_0$$

Now, substitute the definition of $p_{\text{data}}(\mathbf{z}_0|c) = \frac{\gamma(\mathbf{z}_0)p(c|f_\theta(\mathbf{z}_0))}{p(c)}$:

$$p_t(\mathbf{z}, c) = p(c)\int_{\mathbb{R}^m} \mathcal{N}(\mathbf{z}; \alpha_t\mathbf{z}_0, \sigma_t^2 I_m)\frac{\gamma(\mathbf{z}_0)p(c|f_\theta(\mathbf{z}_0))}{p(c)}\mathrm{d}\mathbf{z}_0$$

Assuming $p(c) \neq 0$ (for $\mu_C$-a.e. $c$), we can cancel $p(c)$:

$$p_t(\mathbf{z}, c) = \int_{\mathbb{R}^m} \mathcal{N}(\mathbf{z}; \alpha_t\mathbf{z}_0, \sigma_t^2 I_m)\gamma(\mathbf{z}_0)p(c|f_\theta(\mathbf{z}_0))\mathrm{d}\mathbf{z}_0$$

At $t = 0$, we have $\alpha_0 = 1$ and $\sigma_0 = 0$. The Gaussian density $\mathcal{N}(\mathbf{z}; \alpha_0\mathbf{z}_0, \sigma_0^2 I_m)$ becomes $\mathcal{N}(\mathbf{z}; \mathbf{z}_0, 0 \cdot I_m)$. This is interpreted as the Dirac delta function $\delta(\mathbf{z} - \mathbf{z}_0)$. So,

$$p_0(\mathbf{z}, c) = \int_{\mathbb{R}^m} \delta(\mathbf{z} - \mathbf{z}_0)\gamma(\mathbf{z}_0)p(c|f_\theta(\mathbf{z}_0))\mathrm{d}\mathbf{z}_0$$
$$= \gamma(\mathbf{z})p(c|f_\theta(\mathbf{z})) \quad \text{(by the sifting property of the Dirac delta)}$$

This matches the first target distribution.

At $t = 1$, we have $\alpha_1 = 0$ and $\sigma_1 = 1$. The Gaussian density $\mathcal{N}(\mathbf{z}; \alpha_1\mathbf{z}_0, \sigma_1^2 I_m)$ becomes $\mathcal{N}(\mathbf{z}; 0 \cdot \mathbf{z}_0, 1^2 I_m) = \mathcal{N}(\mathbf{z}; 0, I_m)$. By definition, $\mathcal{N}(\mathbf{z}; 0, I_m) = \gamma(\mathbf{z})$. So,

$$p_1(\mathbf{z}, c) = \int_{\mathbb{R}^m} \gamma(\mathbf{z})\gamma(\mathbf{z}_0)p(c|f_\theta(\mathbf{z}_0))\mathrm{d}\mathbf{z}_0$$

$$= \gamma(\mathbf{z})\int_{\mathbb{R}^m} \gamma(\mathbf{z}_0)p(c|f_\theta(\mathbf{z}_0))\mathrm{d}\mathbf{z}_0$$

The integral $\int_{\mathbb{R}^m} \gamma(\mathbf{z}_0)p(c|f_\theta(\mathbf{z}_0))\mathrm{d}\mathbf{z}_0$ is, by definition, $p(c)$. Therefore,

$$p_1(\mathbf{z}, c) = \gamma(\mathbf{z})p(c)$$

This matches the second target distribution.

Thus, the process defines an interpolation for the joint density $p_t(\mathbf{z}, c)$ between $p_0(\mathbf{z}, c) = \gamma(\mathbf{z})p(c|f_\theta(\mathbf{z}))$ and $p_1(\mathbf{z}, c) = \gamma(\mathbf{z})p(c)$. $\qquad\square$

## A.3 Main Theorem and Proof

**Definition 1** (Interpolation Distribution Sequence (from Lemma 3))**.** *A sequence of time points* $0 = t_0 < t_1 < \cdots < t_N = 1$. *For each* $t_k$, *we have a latent-condition distribution* $\nu_{t_k}$ *(density* $p_{t_k}(\mathbf{z}, \mathbf{c})$*) such that* $\nu_{t_0} = \nu_0$ *and* $\nu_{t_N} = \rho$.

**Theorem A.1.** *Assume the distributions* $\eta, \nu_0, \rho$ *and the interpolation sequence* $\{\nu_{t_k}\}_{k=0}^{N}$ *as defined above. Let* $f_\theta : \mathbb{R}^m \to \mathbb{R}^n$ *be a pre-trained generator, and* $f_{\theta,\phi} : \mathbb{R}^m \times \mathcal{C} \to \mathbb{R}^n$ *be a conditional generator with a single set of trainable parameters* $\phi$. *The map* $T_\phi$ *is defined as* $T_\phi(\mathbf{z}, \mathbf{c}) = (f_{\theta,\phi}(\mathbf{z}, \mathbf{c}), \mathbf{c})$.

*Consider the following two conditions:*

1. **Boundary Condition**: *The parameters* $\phi$ *ensure the boundary loss is zero:*

$$\mathbb{E}_{(\mathbf{z},\mathbf{c}) \sim \nu_{t_0}}[d(f_{\theta,\phi}(\mathbf{z}, \mathbf{c}), f_\theta(\mathbf{z}))] = 0$$

   *where* $d(\cdot, \cdot)$ *is a distance metric on* $\mathbb{R}^n$. *By the Boundary Loss Lemma, this implies* $(T_\phi)_\# \nu_{t_0} = \eta$.

2. **Consistency Condition**: *The parameters* $\phi$ *also satisfy:*

$$L_{total}(\phi) = \sum_{k=0}^{N-1} \mathrm{MMD}^2((T_\phi)_\# \nu_{t_{k+1}}, (T_\phi)_\# \nu_{t_k}) = 0$$

*If such a parameter set* $\phi$ *exists and satisfies both conditions above, then* $f_{\theta,\phi}$ *(when its input is distributed according to* $\rho$*) generates the target data-condition distribution* $\eta$:

$$(T_\phi)_\# \rho = \eta$$

*That is, if* $(\mathbf{z}, \mathbf{c}) \sim \rho$ *(i.e.,* $\mathbf{z} \sim \gamma(\cdot)$ *and independently* $\mathbf{c} \sim p(\cdot)$*), then* $(f_{\theta,\phi}(\mathbf{z}, \mathbf{c}), \mathbf{c}) \sim \eta$ *(i.e., its density is* $p_\theta(\mathbf{x})p(\mathbf{c}|\mathbf{x})$*).*

*Proof.* Let $\phi$ be a parameter set that satisfies the two conditions stated in the theorem.

The first condition is $\mathbb{E}_{(\mathbf{z},\mathbf{c}) \sim \nu_{t_0}}[d(f_{\theta,\phi}(\mathbf{z}, \mathbf{c}), f_\theta(\mathbf{z}))] = 0$. Recall that $\nu_{t_0}$ is the distribution with density $\gamma(\mathbf{z})p(\mathbf{c}|f_\theta(\mathbf{z}))$. According to the Boundary Loss Lemma (Lemma 2), this zero loss implies that the push-forward measure $(T_\phi)_\# \nu_{t_0}$ is equal to the target distribution $\eta$. So, $(T_\phi)_\# \nu_{t_0} = \eta$.

The second condition is $\sum_{k=0}^{N-1} \mathrm{MMD}^2((T_\phi)_\# \nu_{t_{k+1}}, (T_\phi)_\# \nu_{t_k}) = 0$. Since $\mathrm{MMD}^2(P, Q) \geq 0$ for any probability distributions $P, Q$, for the sum of non-negative terms to be zero, each individual term in the sum must be zero. Therefore, for each $k \in \{0, 1, \ldots, N-1\}$:

$$\mathrm{MMD}^2((T_\phi)_\# \nu_{t_{k+1}}, (T_\phi)_\# \nu_{t_k}) = 0$$

Assuming MMD is based on a characteristic kernel, $\mathrm{MMD}^2(P, Q) = 0$ if and only if $P = Q$. Thus, for each $k \in \{0, 1, \ldots, N-1\}$:

$$(T_\phi)_\# \nu_{t_{k+1}} = (T_\phi)_\# \nu_{t_k}$$

The result from step 2 implies a chain of equalities for the push-forward measures generated by $T_\phi$ from the sequence of input distributions $\nu_{t_k}$:

$$\begin{aligned}
(T_\phi)_\# \nu_{t_N} &= (T_\phi)_\# \nu_{t_{N-1}} \\
&= (T_\phi)_\# \nu_{t_{N-2}} \\
&\vdots \\
&= (T_\phi)_\# \nu_{t_1} \\
&= (T_\phi)_\# \nu_{t_0}
\end{aligned}$$

So, we have $(T_\phi)_\# \nu_{t_N} = (T_\phi)_\# \nu_{t_0}$.

From the first condition, we established that $(T_\phi)_\# \nu_{t_0} = \eta$. Substituting this into the equality chain:

$$(T_\phi)_\# \nu_{t_N} = \eta$$

From Lemma A.3, we know that $\nu_{t_N}$ (which corresponds to $p_t(\mathbf{z}, \mathbf{c})$ at $t = t_N = 1$) is the independent latent-condition distribution $\rho$. The density of $\rho$ is $p_\rho(\mathbf{z}, \mathbf{c}) = \gamma(\mathbf{z})p(\mathbf{c})$. Substituting $\nu_{t_N} = \rho$:

$$(T_\phi)_{\#}\rho = \eta$$

This is the desired conclusion. If $(\mathbf{z}, \mathbf{c})$ is sampled from $\rho$ (meaning $\mathbf{z} \sim \gamma(\cdot)$ independently of $\mathbf{c} \sim p(\cdot)$) and then transformed by $T_\phi$ (i.e., forming $(f_{\theta,\phi}(\mathbf{z}, \mathbf{c}), \mathbf{c})$), the resulting distribution is the target data-condition distribution $\eta$ (which has density $p_\theta(\mathbf{x})p(\mathbf{c}|\mathbf{x})$).

$\square$

**Remark 1** (Lemma 2 in main paper)**.** *Specifically, when we take $N = 1$ (particle number) in MMD loss, and take we have some specific kernel choice:*

- *$k(x, y) = -\|x - y\|^2$, although it is not a proper positive definite kernel required by MMD, we find it works well in practice*

- *$k(x, y) = \mathbf{c} - \sqrt{\|x - y\|^2 + \mathbf{c}^2}$ is a conditionally positive definite kernel.*

*Then the summed MMD Loss*

$$L_{total}(\phi) = \sum_{k=0}^{N-1} \mathrm{MMD}^2((T_\phi)_{\#}\nu_{t_{k+1}}, (T_\phi)_{\#}\nu_{t_k}) = 0,$$

*can be implemented in a practical way:*

$$L_{total}(\phi) = \sum_{k=0}^{N-1} \mathbb{E}_{\gamma(\mathbf{z})p(\mathbf{c}|f_\theta(\mathbf{z}))}\mathbb{E}_{\gamma(\mathbf{w})} d(f_{\theta,\phi}(\alpha_{t_{k+1}}\mathbf{z} + \sigma_{t_{k+1}}w, \mathbf{c}), f_{\theta,\phi}(\alpha_{t_k}\mathbf{z} + \sigma_{t_k}w, \mathbf{c})),$$

*where $d$ is $l_2$ loss or pseudo-huber loss, other kernel-induced losses also work.*

## B  Experiment Details

**One-step generator**  We adopt Diff-Instruct [3] for pre-training the one-step generator. We adopt the AdamW optimizer. The $\beta_1$ is set to be 0, and the $\beta_2$ is set to be 0.95. The learning rate for the generator is $2e - 6$, the learning rate for fake score is $1e - 5$. We apply gradient norm clipping with a value of 1.0 for both the generator and fake score. We use batch size of 256.

**Controllable Generation**  We use Contorlnet's architecture [15] for training. We adopt the AdamW optimizer with $\beta_1 = 0.9$, $\beta_2 = 0.95$, and the learning rate of $1e - 5$. We use batch size of 128.

**Image-prompted Geneartion**  We use IP-adapter's architecture [39] for training. We adopt the AdamW optimizer with $\beta_1 = 0.9$, $\beta_2 = 0.95$, and the learning rate of $1e - 4$. We use batch size of 128. We use a probability of 0.05 to drop text during training.

## C  Limitations

Our model shares common challenges with other controllable text-to-image diffusion models, particularly regarding fairness considerations and precise detail handling. We plan to explore these ongoing challenges in the generation domain in our future works, to improve the model's performance in text synthesis, fairness, and fine-grained control.

## D  Broader Impacts

This work presents NCT, a method that can inject new controls into pre-trained one-step generators. From a positive perspective, this academic contribution has potential for widespread industrial adoption, where its computational efficiency could reduce energy consumption and provide environmental advantages. Conversely, malicious use of such rapid generation technologies could facilitate the faster production of harmful content. While our focus remains on scientific advancement, we are committed to mitigating risks through measures such as removing harmful material from training datasets.

# E   Safeguards

The NCT is trained on an internally curated dataset that has undergone rigorous human and machine-based filtering to exclude harmful or violent content.

