# OpenReview forum: "Noise Consistency Training: A Native Approach for One-step Generator in Learning Additional Controls"
_NeurIPS.cc/2025/Conference — NeurIPS 2025 poster_

### Official Review · Reviewer_92Eo · 2025-06-26

**Clarity:** 2
**Significance:** 2
**Originality:** 2
**Rating:** 3
**Confidence:** 5

**Summary:**

This paper introduces Noise Consistency Training (NCT), a lightweight method to adapt pre-trained one-step generators for controllable content generation without retraining the base model or requiring original training data. NCT employs an adapter module and a noise consistency loss to align generation behavior with new control signals. Essentially, the proposed method combines the Consistency model and ControlNet. Authors evaluated NCT based on pretrained models such as SD on COCO benchmarks.

**Questions:**

As in weaknesses

**Ethical Concerns:**

["NO or VERY MINOR ethics concerns only"]

**Final Justification:**

Thanks for the detailed reply. It resolves some of my concerns but not all of them. The notation problems and the concerns about empirical results are solved. However, the resemblance with Consistency model and ControlNet still constrain the contribution of this work. After thorough consideration, I decide to raise my score to 3 and not against for acceptance.

**Limitations:**

No analysis of when or why NCT might fail is presented.

**Quality:**

2

**Strengths And Weaknesses:**

### **Strengths**

1.  **Practical Significance & Well-Defined Problem:** The paper tackles a highly relevant and practical challenge in AIGC: efficiently adapting powerful one-step generators to diverse control signals without costly retraining or distillation. The focus on computational efficiency and deployability addresses a key bottleneck in real-world applications.
2.  **Modularity and Data Efficiency:** The proposed NCT framework is commendably lightweight and modular. Its reliance solely on the pre-trained generator and a control signal model, without needing original training data or full model retraining, enhances its practical utility and reduces deployment barriers.

### **Weaknesses**

1.  **Technical Writing and Presentation:**
    *   **Formula 1 (ControlNet Representation):** The overly simplistic reduction of ControlNet's complex architecture to a single parameter `phi` is misleading and does not accurately reflect its known structure or contributions. This requires substantial revision and proper explanation.
    *   **Figure 1:** The figure lacks meaningful discussion or analysis within the main text. Its relevance to NCT's mechanism or results is unclear, making it appear disconnected from the core narrative. A detailed caption and textual integration are essential.
    *   **Formula 3 (Definition of `d`):** The variable `d` is used without a clear definition. Its meaning and role in the loss function must be explicitly stated.
    *   **Formula 6 (Notation & Errors):**
        *   The presence of `#Simplify Notation` is unacceptable in a final manuscript and suggests unrevised drafting artifacts.
        *   The chaining of three expectation operators (`E`) is unnecessarily complex and hinders readability without a clear justification. Simplification or splitting is advised.
        *   The subscript `z,c | z,ε` appears erroneous and logically inconsistent (`z` appears conditioned on itself?). The intended conditional variables must be corrected and precisely defined.
    *   **Formula 13 (Norm Symbol):** The norm symbol (`|| · ||_1`) lacks specification (e.g., L1, L2). The choice of norm can impact the loss behavior and results; this must be clarified.


2.  **Limited Novelty & Insufficient Contextualization (Major Concern):**
    *   The core innovation, NCT, appears heavily reliant on applying a consistency loss principle (well-established in consistency models) to the specific context of adapting ControlNet-like structures for one-step generators. While the *combination* is novel and practically useful, the paper fails to sufficiently differentiate the *fundamental contribution* beyond this synergistic application.
    *   The discussion lacks depth in contrasting NCT with highly relevant alternative approaches for efficient adaptation or conditional generation, particularly:
        *   **Shortcut Learning/Modeling:** How does NCT compare conceptually and empirically to methods designed for fast adaptation with minimal changes?
        *   **Meanflow / Other Recent Efficient Generators:** The absence of comparison or discussion regarding very recent advances like Meanflow, which also target efficient high-quality generation (potentially including controllability aspects), weakens the claim of state-of-the-art positioning and novelty. A thorough related work update is needed.
    *   **Risk:** The work risks being perceived primarily as a competent engineering solution ("experimental report") applying known concepts (consistency, adapters) to a new task, rather than introducing a fundamentally new principle or significant theoretical advancement.

3.  **Superficial and Problematic Theoretical Analysis:**
    *   **Derivative Nature:** Lemmas and theorems strongly resemble those from the foundational consistency model literature without clear novel theoretical extension or significant adaptation specific to NCT's conditional control objective. The paper does not convincingly establish what *new theoretical insight* NCT provides.
    *   **Lack of Cohesive Narrative:** The theoretical components feel bolted on rather than integral. There's a disconnect between the listed lemmas/theorems and the core arguments or practical results of the paper. A stronger narrative linking theory to the proposed method and its empirical success is missing.
    *   **Confusing Claims:** Key arguments lack supporting evidence or rigorous justification:
        *   Line 123: Attributing blurry images solely to "high variance of the optimized objective" is asserted without analysis, proof, or citation. What variance? Compared to what?
        *   Line 130: The claim about the model's inability to perform conditional generation given random `z` due to training on coupled pairs `(z, c)` is stated dogmatically. No theoretical argument or experimental evidence (e.g., showing failure cases) is provided.


4.  **Insufficient Experimental Scope and Depth:**
    *   **Limited Resolution:** Heavy reliance on low-resolution image datasets significantly diminishes the claim of "high-quality" generation and practical relevance. Modern AIGC demands robust performance at higher resolutions. Results on standard high-resolution benchmarks are crucial.
    *   **Narrow Condition Scope:** Experiments focus on a limited set of control signals. The paper does not demonstrate generality across diverse, complex, or semantically rich conditions. How broadly applicable is NCT really?
    *   **Lack of Ablation Studies:** The current ablation studies are overly simple. Critical components lack ablation:
        *   How essential is the specific adapter architecture?
        *   What is the impact of the noise consistency loss vs. simpler losses?
        *   How does performance scale with the amount of adaptation data?

[1] One Step Diffusion via Shortcut Models
[2] Mean Flows for One-step Generative Modeling

---

> ### Author Rebuttal · Authors · 2025-07-31
>
> We sincerely thank you for your time and effort in reviewing our paper. We address each comment below.
>
> > The overly simplistic reduction of ControlNet's architecture to a single parameter does not accurately reflect its structure/contributions.
>
> **Our NCT is agnostic to the specific structure of the ControlNet or adapter**.
> The notation $\phi$ is used as a general representation of the adapter's learnable parameters, not as a reduction of its complexity. Whether the adapter is a standard ControlNet, a lightweight adapter like IP-Adapter, or any other control module, our training methodology applies uniformly.
> **Our work is not in modifying ControlNet's architecture, but in developing an effective training strategy for *any* control adapter when applied to one-step generators.** We chose this general notation precisely because our method works across different adapter architectures without requiring architecture-specific modifications.
>
> > Figure 1 lacks discussion or analysis in main text. Its relevance to NCT's mechanism or results is unclear.
>
> Figure 1 is the overall framework description of NCT with a clear caption and contents.
> We will provide a brief introduction of Figure 1 in the method section to make it clearer.
>
> > The chaining of three expectation operators is unnecessarily complex and hinders readability.
>
> We have simplified the notation in the second line of Eq. (6).
>
> > The presence of \#Simplify Notation is unacceptable and suggests unrevised drafting artifacts.
>
> The "\#Simplify Notation" is to clearly indicate that we are simplifying notation in Line 2 from Line 1 of Eq. (6).
>
> > The subscript $z,c|z,\epsilon$ appears erroneous and logically inconsistent (z appears conditioned on itself?).
>
> This denotes three variables rather than one variable, where each is separated by ",".
>
> > The symbol ($|| \cdot ||_1$) lacks specification (e.g., L1, L2).
>
> We note that the symbol $||\cdot||_1$ exactly denotes the L1 norm.
>
> > The core innovation, NCT, appears heavily reliant on applying a consistency loss principle (well-established in consistency models) to the specific context of adapting ControlNet-like structures for one-step generators. While the combination is novel and practically useful, the paper fails to sufficiently differentiate the fundamental contribution beyond this synergistic application.
>
> **NCT is not a combination of Consistency Training (CT) and ControlNet.**
> There is a fundamental and significant difference between Noise Consistency Training (NCT) and CT. NCT "diffuses" noise in the noise space and "denoises" the noise to learn adapters for adding controls to one-step generators, while CT adds noise to data in the data space and denoises the noisy data to learn few-step generator.
>
> > The discussion lacks depth in contrasting NCT with relevant alternative approaches for efficient adaptation or conditional generation (Shortcut Models and Meanflows).
>
> **NCT is orthogonal to the specific method used for learning the one-step generator**. Since NCT focuses on injecting conditions into a well-trained, one-step generator, which is agnostic to the method used to train the one-step generator.
> Furthermore, current approaches like Shortcut Models and Meanflows do not prioritize controllable generation, nor have they demonstrated the capability for high-quality, one-step text-to-image synthesis. In contrast, NCT is capable of achieving high-quality, controllable text-to-image generation in one step.
>
> > Lemmas\&theorems strongly resemble those from the foundational consistency model literature without clear significant adaptation specific to NCT's conditional control objective. The paper does not convincingly establish what new theoretical insight NCT provides.
>
> We respectfully argue that NCT is not a derivative work but a novel approach with a fundamental distinct theoretical and algorithmic foundation.
>
> **The core difference is that NCT enforces consistency in the noise space, while Consistency Models (CMs) operate in the data space**. These Lemmas\&theorems are necessary reformulation, not a minor adaptation. In particular, NCT "diffuses" noise in the noise space and "denoises" the noise, while CMs adds noise to data in the data space and denoises the noisy data. NCT is designed for pre-trained one-step generators to add controls, which lack the multi-step PF-ODE trajectory originating from a clean data point  that CMs fundamentally rely on.
>
> This algorithmic shift necessitates a different theoretical underpinning. While CM theory is rooted in approximating ODE solutions, NCT's theory is based on distribution matching. Our noise diffusion process creates an interpolation path from a coupled to an independent latent-condition distribution. The NCT objective, including the crucial boundary loss, forces the model's output distribution to remain invariant along this path.
>
> > There's a disconnect between the listed lemmas/theorems and the core arguments or practical results of the paper. A stronger narrative linking theory to the proposed method and its empirical success is missing.
>
> We will revise the manuscript to make the connection between our theory and practice more explicit. The narrative is as follows:
>
> Our main theorem establishes the ideal objective. It proves that a generator can learn the true conditional posterior if it satisfies two abstract conditions: a Boundary Condition and a Consistency Condition.
>
> Our proposed method is a direct, practical implementation of this theory. The theoretical Boundary Condition is directly optimized by our Boundary Loss ($L_{bound}$).
> The theoretical Consistency Condition (formalized as an MMD objective in Lemma is empirically approximated by our Noise Consistency Loss ($L_{con}$).
>
> The strong empirical results are the validation of this framework. They demonstrate that our practical losses effectively serve as proxies for the theoretical ideals, successfully guiding the model to the desired conditional distribution. Our ablation study, which shows complete model failure when either loss is removed, further reinforces that both theoretical pillars (and their practical counterparts) are essential.
>
> > Attributing blurry images solely to "high variance of the optimized objective" is asserted without analysis, proof, or citation. What variance? Compared to what?
>
> The "high variance" refers to the variance in the conditional distribution $p(x|x_T)$ when optimizing the direct denoising objective. When $x_T$ is pure noise, there are infinitely many possible clean images $x$ that could have generated it, making $p(x|x_T)$ highly multimodal. The optimal solution for minmizing the direct denoising loss under an L2 loss is $\mathbb{E}[x|x_T]$, which averages over all possible images in the dataset, inherently producing blurry results.
>
> To substantiate this claim, we conducted an empirical analysis measuring the variance of the optimization objectives:
>
> | Method           | Variance |
> | ---------------- | -------- |
> | Direct denoising | 0.956|
> | NCT| 0.013|
>
> The results clearly demonstrate that the noise consistency loss exhibits significantly lower variance compared to direct denoising, supporting our assertion.
>
> > The claim about the model's inability to perform conditional generation given random z due to training on coupled pairs (z, c) is stated dogmatically. No theoretical argument or experimental evidence is provided.
>
> **We have ablated this variant in our main paper, showing that it is unable to handle the condition**. In particular, training on coupled pairs (z, c) is equal to our ablated variant "w/o noise consistency training". Results are in Figure 3 and Table 4, indicating that this approach is unable to inject conditions.
>
> > Limited Resolution
>
> Our experiments are conducted over 512 resolution, which is already relatively high-resolution and widely adopted in most prior works in the line of distillation [3, 5, 9] and also the original multi-step ControlNet [12].
>
> > Narrow Condition Scope.
>
> **Our experiments focused on controllable generation (based on ControlNet) and Image-prompted generation (IP-Adapter). We believe this demonstrates NCT's broad applicability rather than limiting it.**
>
> Our experimental design deliberately selected these two architectures because they represent fundamentally different conditioning paradigms: ControlNet exemplifies spatial/structural control (edges, depth, HED, low-resolution) while IP-Adapter represents semantic/style control through image prompts. Together, they span the spectrum from low-level geometric constraints to high-level semantic guidance.
>
> Moreover, ControlNet and IP-Adapter serve as foundational architectures for numerous downstream applications. We believe that NCT's experiments have covered a sufficient number of conditional scenarios.
>
> > How essential is the specific adapter architecture?
>
> **Our NCT is agonistic to specific adapter architecture, since NCT is an algorithm in training adapters to add new controls for one-step generators.** We apply the same adapter architecture among methods in our experiments. NCT can effectively train adapters for one-step generator in adding new controls, as indicated by the success on both ControlNet and IP-Adapter architecture.
>
> > What is the impact of the noise consistency loss vs. simpler losses?
>
> We have ablated that simply use boundary loss without noise consistency loss in Table 3 and Figure 4. It can be seen that without noise consistency loss, it is unable to inject conditions. Additionally, we conduct an extra ablation study on simply using direct denoising loss:
> | Method                     | FID↓  | Con.↓ |
> | -------------------------- | ----- | ----- |
> | Ours| 13.67 | 0.110 |
> | w/o noise consistency loss | 20.56 | 0.165 |
> | w/ simple direct denoising | 37.49 | 0.115 |
> The results show that the simple denoising loss leads to inferior generation, while our full approach can ensure high-quality generation and condition alignment.

---

> > ### Comment · Reviewer_92Eo · 2025-08-04
> >
> > Thanks for the detailed reply. It resolves some of my concerns but not all of them. I will update the score accordingly.

---

> > > ### Author Response · Authors · 2025-08-05
> > >
> > > Thanks for your reply. We are happy that our response addresses some of your concerns. May you please tell us the remaining concerns? We are glad to provide further clarification to address the remaining concerns.

---

### Official Review · Reviewer_X1xZ · 2025-06-27

**Clarity:** 1
**Significance:** 3
**Originality:** 3
**Rating:** 3
**Confidence:** 3

**Summary:**

The paper introduces Noise Consistency Training (NCT), a lightweight procedure for adding new control signals to a pre-trained one-step generator without re-running diffusion distillation or accessing the original training set. NCT diffuses latent noise rather than images, forming pairs of latents with different noise levels; an adapter is trained with a noise-space consistency loss that pushes its outputs for the two latents to match, plus a boundary loss that keeps the adapter anchored to the original generator.  The proposed method is highly modular, data-efficient, and easy to deploy.

**Questions:**

See "weaknesses".

**Ethical Concerns:**

["NO or VERY MINOR ethics concerns only"]

**Final Justification:**

We appreciate the authors' effort in the rebuttal stage. However, my primary concern lies in the writing quality and mathematical rigor. For example, Lemma 1 does not explicitly state all the assumptions used in its proof. In addition, the paper has some unclear aspects in notation and presentation, which I feel are not minor and may require substantial revision beyond the scope of the rebuttal. Therefore, I will keep my score (3), but I will lower my confidence from 4 to 3.

**Limitations:**

yes

**Paper Formatting Concerns:**

No concern.

**Quality:**

3

**Strengths And Weaknesses:**

Strengths:

1. The paper introduces Noise Consistency Training (NCT), a conceptually simple yet novel approach that enables controllable image generation using a one-step generator. It trains a lightweight adapter using a consistency loss in the latent noise space, effectively sidestepping the need for teacher models or expensive multi-step diffusion processes.

2. Empirical results demonstrate that NCT achieves a significant reduction in computational cost (from 50 NFEs to just 1 NFE) while maintaining or even improving generation fidelity and control alignment, across multiple control types.

3. The paper includes ablation experiments showing that both the boundary loss and noise consistency loss are necessary to achieve good performance, demonstrating the value of these components in training stable and condition-sensitive models.

Weakness:
1. The paper is generally not well written and can be difficult to follow, especially for readers who are not already familiar with the  consistency models. For instance, the presentation of consistency models and Equation (4), which defines the consistency loss, lacks clarity. Could the authors provide a clearer explanation of why this objective can force inject conditioning signals, and definition of the stop-gradient operator?
 Moreover, the paper suffers from inconsistent and overloaded notation: time steps $t_n$, $t_k$  are used ambiguously as subscripts for both $x$ and $z$. and the definitions of diffusion parameters in later equations (e.g., (5), (6)) are inconsistent with those used earlier.


2. The training procedure assumes sampling from $p(c \mid f_\theta(z))$, and the paper appears to treat this step as a black-box oracle without further justification or clarification. However, for conditions like Canny, HED, or depth maps, such conditional distributions are not readily defined or tractable to sample from to me. Could the authors elaborate on how this sampling is performed in practice?


3. Lemma 1 makes its claim without stating the necessary assumptions in the main text. While the appendix provides a specific parameter scheduling under which the result holds ($\alpha_0 = 1, \sigma_0 = 0$ and $\alpha_1 = 0, \sigma_1 = 1$), this renders the lemma rather trivial. However, the main paper does not clearly state this parameter scheduling.


4. The paper repeatedly asserts that the proposed noise consistency loss yields lower variance, yet it offers neither theoretical justification  nor empirical evidence to support this claim. I believe this is true, but could the authors give a more precise explanation?

---

> ### Author Rebuttal · Authors · 2025-07-31
>
> We sincerely thank you for your time and effort in reviewing our paper. We address each comment below.
>
> > The paper can be difficult to follow for readers who are not already familiar with the consistency models.  For instance, the presentation of consistency models and Equation (4), which defines the consistency loss, lacks clarity.
>
> Thank you for your valuable feedback.
> We will expand the preliminary section to include a comprehensive introduction to consistency models, ensuring readers unfamiliar with this framework can follow our work more easily.
> Besides, while consistency models serve as inspiration for our noise consistency training, we want to clarify that our approach is fundamentally self-contained. The key differences include: 1) our noise consistency training operates in a different space and uses a distinct formulation; 2) the theoretical foundations and practical implementation differ significantly from standard consistency models.
>
> > Could the authors provide a clearer explanation of why consistency training can force inject conditioning signals, and definition of the stop-gradient operator?
>
> For consistency training with condition $c$, it tries to make $g(x_{t_{n+1}}, c)$ approximate $g(x_{t_{n}}, c)$, where $t_{n+1} > t_n$ and  $g$ is the desired consistency model, taking noisy samples with condition and predicts clean samples.
> Intuitively, the network can better predict clean samples given less noisy samples $x_{t_{n}}$, hence the $g(x_{t_{n}}, c)$ can be used as the target for learning $g(x_{t_{n+1}}, c)$.
> In this case, the information in $x_{t_{n+1}}$ about the image is more corrupted than $x_{t_{n}}$, which can force the network to utilize the information from condition $c$ to minimize the loss.
>
> The stop-gradient operator treats a variable as a constant, preventing gradients from backpropagating through it.
>
> > Moreover, the paper suffers from inconsistent and overloaded notation timesteps $t_n, t_k$, are used ambiguously as subscripts for both $x$ and $z$.
>
> Sorry for the confusion. We will clean up the notation in our revision.
>
> > The definitions of diffusion parameters in later equations (e.g., (5), (6)) are inconsistent with those used earlier.
>
> We deliberately use different symbols to distinguish between different models. Specifically, Eq. 4 is a standard consistency model ($g_\alpha$), while Eq. 6 involves a one-step generator ($f_{\theta}$) and its adapted version ($f_{\theta,\phi}$).
>
> > The definition of $p(c|x)$
>
> We model the distribution $p(c|x)$ as a Dirac delta distribution: $p(c|x) \triangleq \delta(c - h(x))$, where $h(\cdot)$ represents a deterministic discriminative model that maps inputs to conditions. The Dirac delta distribution can be sampled easily by $h(x)$, but it is hard to estimate the density. This is well-suited for our NCT, since NCT just requires samples from $p(c|x)$ for training.
>
> > Lemma 1 makes its claim without stating the necessary assumptions in the main text. While the appendix provides a specific parameter scheduling under which the result holds this renders the lemma rather trivial. However, the main paper does not clearly state this parameter scheduling.
>
> We are sorry for the confusion on the scheduler.
> We note that our schedule is specified by $z_t = \sigma_t z + \sqrt{1-\sigma_t}\epsilon$. And we acknowledge a typo in Eq. 5, where the $z_T=\sigma_t z + \sqrt{1-\sigma_t}\epsilon$ should be $z_t=\sigma_t z + \sqrt{1-\sigma_t}\epsilon$.
> Besides, $z_T$ denotes the terminal diffused noise (i.e., $\sigma$ = 1), while $z_0$ denotes the initial noise (i.e., $\sigma$ = 0). This is a widely used notation in diffusion literature, and we will clarify this in the revision.
>
> Regarding Lemma 1, it is given for \textit{formally stating} that our diffusion process can obtain independent noise-condition pairs, which is crucial for enabling our approach to learn conditional generation.
> Although the approach is straightforward, it provides a formal ground for subsequent analysis.
>
> > The paper repeatedly asserts that the proposed noise consistency loss yields lower variance, yet it offers neither theoretical justification nor empirical evidence to support this claim. I believe this is true, but could the authors give a more precise explanation?
>
> The lower variance is compared to the direct denoising objective. For the direct denoising objective, the target samples follow conditional distribution $p(x|x_T)$ which is highly multimodal. Since when $x_T$ is pure noise, there are infinitely many possible clean images $x$ that could have generated it.
> In contrast, for noise consistency loss, the target samples follows $q(x|z_{t_{n}},c) = \int p(z_{t_{n}}|z_{t_{n+1}})p(x|z_{t_n},c)$, where $p(x|z_{t_n},c)\triangleq \delta(x-f(z_{t_n},c))$.
> And $p(z_{t_{n}}|z_{t_{n+1}})$ has small variance, since the timestep gap between $z_{t_{n+1}}$ and $z_{t_{n}}$ is chosen to be small.
> Based on the above intuition, we believe our method produces lower variance.
>
> To substantiate this claim, we conducted an empirical analysis measuring the variance of the optimization objectives:
>
> | Method           | Variance |
> | ---------------- | -------- |
> | Direct denoising | 0.956    |
> | NCT              | 0.013    |
>
> The results clearly demonstrate that the noise consistency loss exhibits significantly lower variance compared to direct denoising, supporting our assertion.

---

> > ### Comment · Reviewer_X1xZ · 2025-08-06
> >
> > Thank you for the rebuttal and the clarifications. However, my primary concern about writing quality and mathematical rigor remains unresolved, and I believe the paper is still not ready for publication in its current form. In particular, all the settings and assumptions are not fully stated in Lemma 1. The result also relies on a very specific scheduler, which makes the theoretical result trivial. In addition, the purpose and significance of Lemma 1 and 2 are not clearly explained.  In addition, the paper has some unclear aspects in notation and presentation, which I feel are not minor and may require substantial revision beyond the scope of the rebuttal. For these reasons, I will keep my score (weak reject), but I will lower my confidence.

---

> > > ### Author Response · Authors · 2025-08-06
> > >
> > > Thank you for your reply. We would like to provide further clarification below:
> > >
> > > We note that ***our main theoretical result is Theorem 1, which formally establishes that our approach, in theory, empowers the one-step generator to incorporate additional controls***.
> > > Before stating Theorem 1, we first introduce Lemma 1 and Lemma 2, serving as preliminary steps to Theorem 1, for the reader's convenience rather than being merged into Theorem 1.
> > > In particular:
> > >
> > > - Lemma 1 is for formally stating the property of our defined noise diffusion process.  Although the proof of Lemma 1 is rather simple and straightforward, it further highlights the elegance of our approach.
> > > - Lemma 2 builds the connection between our noise consistency loss and a distribution-level loss between conditional distributions.
> > >
> > > We would like to highlight that our central contribution and novelty lies in the proposed noise consistency training, which is ***the first algorithm that enables adding controls to the one-step generator without relying on extra diffusion distillation or retraining the base diffusion model***. We believe this denotes a valuable contribution to the NeurIPS community.
> > >
> > > We hope to further clarify that the Consistency Models (CMs) in our paper are only the background in diffusion distillation. Readers do not need to understand CMs to understand our NCT, since both the formulation and theoretical support of our NCT are essentially different from CMs.
> > > We believe our introduction of our own method is self-contained.
> > >
> > > Nevertheless, we will try our best to address the raised issues in our revision.
> > > We feel confident that we can make the writing and presentation clear enough to the general readers without substantial structural changes.

---

### Official Review · Reviewer_1vDf · 2025-06-30

**Clarity:** 3
**Significance:** 3
**Originality:** 3
**Rating:** 5
**Confidence:** 4

**Summary:**

This paper addresses the significant challenge of efficiently integrating new control signals into pre-trained one-step generation models. Existing approaches have had the disadvantage of high computational costs, requiring modifications to the base model and performing additional diffusion distillation. To solve these problems, the authors propose Noise Consistency Training (NCT), a novel and lightweight approach. NCT operates by adding adapter modules to pre-trained one-step generators without requiring retraining of the original training data or the base diffusion model. The key idea is to define a 'noise consistency loss' in the generator's noise space. This loss function ensures that the adapted model's generation results are consistent across noises with different conditional dependencies, thereby implicitly learning new control signals. Additionally, they use a boundary loss alongside this to stabilize training and prevent generated images from deviating significantly from the data distribution. The authors claim to experimentally demonstrate that NCT is modular and data-efficient, achieving state-of-the-art controllable generation with only a single forward pass, surpassing existing multi-step and distillation-based methods in both quality and efficiency.

**Questions:**

1. On a Fairer Comparison with ControlNet: The proposed NCT methodology appears to be applicable not only to one-step generators but also to standard multi-step diffusion models. To provide a more direct and fair comparison, could you apply NCT on a standard diffusion model (e.g., Stable Diffusion 1.5) and benchmark it against the original ControlNet using the exact same base model and evaluation settings (e.g., 50 NFEs)? The current comparisons in Table 1 are challenging to interpret on a one-to-one basis, as they compare models with different base architectures and step counts. This new experiment would help isolate the performance of the training methodology itself.

2. On the Lack of Qualitative Comparisons: The paper relies heavily on quantitative metrics but lacks crucial qualitative comparisons against the most relevant competing methods. While quantitative scores are important, they do not fully capture perceptual quality, the subtlety of control adherence, or potential failure modes. Could you please provide side-by-side visual comparisons of your method's outputs against those from other one-step controllable generation models, such as JDM? Such qualitative evidence is necessary to substantiate the claim that your method achieves state-of-the-art performance.

3. On the Generalizability of NCT: The experiments currently focus on structural control signals, such as Canny edges and depth maps. Have you explored the efficacy of NCT with non-structural or stylistic control signals, for instance, controlling for a specific texture or an artistic style? Could you elaborate on how the core assumption of NCT—that diffusing noise progressively decouples it from the condition—is expected to hold for features that are less spatially robust? For such controls, even minor perturbations in the noise space might lead to significant semantic deviations, potentially challenging the consistency objective.

**Ethical Concerns:**

["NO or VERY MINOR ethics concerns only"]

**Final Justification:**

The author well addressed the concerns. So my intention is clear accept.

**Limitations:**

yes

**Paper Formatting Concerns:**

.

**Quality:**

3

**Strengths And Weaknesses:**

## Strengths

**Originality and Significance** : This paper addresses a highly important and practical problem in the field of Artificial Intelligence Generated Content (AIGC). Utilizing efficient one-step generators, born from the distillation of diffusion models, with additional controls is essential for many applications. The proposed Noise Consistency Training (NCT) method is original in that it performs consistency learning directly in the noise space, unlike conventional consistency models that operate in the image space. This approach offers significant academic contributions by presenting a new avenue for adding controls directly to pre-trained models, thereby avoiding expensive retraining or distillation processes.





**Clarity** : The paper is well-written and clear overall. The problem definition, proposed method, and experimental design are logically well-connected. In particular, the core idea of NCT—progressively decoupling the condition by diffusing the noise and optimizing for consistency between generation results from different noise levels—is intuitive and easy to grasp. The process of introducing a primal-dual algorithm to solve the constrained optimization problem is also clearly explained.






**Quality (Empirical Results)** : The proposed method demonstrates strong performance both quantitatively and qualitatively. In Table 1, NCT achieves comparable or superior FID (image quality) and consistency scores with just 1 NFE (Number of Function Evaluations), compared to the standard ControlNet which requires 50 NFEs. This signifies a dramatic improvement in computational efficiency while maintaining high control quality. The qualitative comparison in Figure 2 also shows that while a conventional approach (DI+ControlNet) suffers from image quality degradation when integrating control signals, NCT successfully incorporates the control while maintaining high fidelity.





## Weaknesses

**Quality (Scope and Generalizability)** : The paper's experiments are limited to structural controls, such as Canny edges, HED, and depth maps. The core assumption of NCT—that diffusing the noise progressively decouples it from the condition —may not hold for spatially less robust features like texture, style, or other fine-grained attributes. For instance, when using an artistic style as a control signal, minor changes in the noise could lead to significant stylistic shifts, potentially making the consistency training ineffective. Although the paper claims general applicability, the experimental evidence provided is insufficient to fully support this.

**Inappropriate Baseline Selection** : A direct comparison with JDM is omitted in Table 2. While the authors argue that JDM is inefficient due to its need for additional distillation, a direct comparison of the final model's performance (in terms of FID and Consistency) is essential to properly situate NCT, even acknowledging the efficiency difference. Furthermore, the paper compares against IP-Adapter in Table 2 but not in Table 1, giving the impression that baselines were selectively chosen for each task, potentially to present the results more favorably.

**Insufficient Ablation Study** : In Section 3.1, the paper claims that a direct denoising approach yields "blurry images," and while Table 3 shows a very high FID for the "w/o noise consistency loss" ablation, it fails to provide a visual example, unlike the other ablations in Figure 4. This makes the claim less convincing, as it is difficult to assess the degree and nature of the "blurriness" based solely on the FID score.



**Question on the Generality of the Method** : A key question is why this method is presented as specific to one-step/consistency-based models. The proposed training scheme seems general. Could the NCT framework be used as an alternative training objective for adapters on standard, multi-step diffusion models like ControlNet? A discussion on this point would help clarify the fundamental contributions and limitations of the proposed loss functions.

---

> ### Author Rebuttal · Authors · 2025-07-31
>
> We sincerely thank you for your time and effort in reviewing our paper. We address each comment below.
>
> > Insufficient Ablation Study : In Section 3.1, the paper claims that a direct denoising approach yields "blurry images," and while Table 3 shows a very high FID for the "w/o noise consistency loss" ablation, it fails to provide a visual example, unlike the other ablations in Figure 4. This makes the claim less convincing, as it is difficult to assess the degree and nature of the "blurriness" based solely on the FID score.
>
> The results for the 'w/o noise consistency' variant are presented in both Figure 4 and Table 3.
> This variant is distinct from direct denoising, for which we report the results below:
>
> | Method              | FID↓  | Con.↓ |
> | ------------------- | ----- | ----- |
> | Ours                | 13.67 | 0.110 |
> | w/ direct denoising | 37.49 | 0.115 |
>
> It can be seen that direct denoising shows much worse FID due to its blurry samples. For visual samples, we will provide them in the revision. We cannot provide it in the rebuttal stage due to the policy. However, we refer to Figure 3 of JDM, which shows results of direct denosing with an additional distillation loss (Diff-instruct loss). It can be seen that the direct denoising approach still produces blurry samples even with an additional distillation loss.
>
> It is worth noting that the reason why the variant (w/o noise consistency loss) exhibits a worse FID score compared to NCT is that the FID calculation is performed between the real images and the samples generated under the conditions corresponding to the same real images.
> Controllable generation with $c\sim p(c|x)$ allows for a distribution closer to that of the real image $x$.
>
>
>
> > The paper compares against IP-Adapter in Table 2 but not in Table 1
>
> This is because the IP-Adapter in Table 2 and ControlNet in Table 1 focus on different tasks. ControlNet is concerned with controllable generation, aiming to preserve the details of the given conditions. In contrast, the IP-Adapter focuses on image-prompted generation, where the main goal is to inject the semantics and style of a reference image. This makes it unsuitable to evaluate them using the same benchmark.
>
> > Compare JDM regarding image-prompted generation
>
> Great suggestion. We compare JDM regarding image-prompted generation below:
>
> | Method      | NFE↓ | Clip-T↑ | Clip-I↑ |
> | ----------- | ---- | ------- | ------- |
> | IP-Adapter | 100  | 0.588   | 0.828   |
> | JDM         | 1    | 0.585   | 0.826   |
> | Ours        | 1    | 0.593   | 0.821   |
>
> It can be seen that NCT achieves competitive results compared to JDM, which requires additional expensive distillation loss.
>
> > On a Fairer Comparison with ControlNet: The proposed NCT methodology appears to be applicable not only to one-step generators but also to standard multi-step diffusion models. To provide a more direct and fair comparison, could you apply NCT on a standard diffusion model (e.g., Stable Diffusion 1.5) and benchmark it against the original ControlNet using the exact same base model and evaluation settings (e.g., 50 NFEs)? The current comparisons in Table 1 are challenging to interpret on a one-to-one basis, as they compare models with different base architectures and step counts. This new experiment would help isolate the performance of the training methodology itself.
>
> **We first clarify that the network architecture (Controlnet + UNet) adopted by NCT is identical to that of standard ControlNet.**
> The difference is that NCT trains a ControlNet for a one-step generator distilled from a pre-trained diffusion model, whereas standard ControlNet trains a ControlNet for the diffusion model itself.
> ***NCT actually faces a greater challenge because it needs to inject the condition in one-step generation.***
> Hence, we believe the comparison is fair for baselines and NCT's advantage is convincing.
>
> NCT is tailored for one-step generators, and extending it to standard diffusion models presents some difficulties.
> The reason is: NCT training requires access to the model's predicted $x_0$ and its corresponding condition $c$. This is not feasible for diffusion models because their one-step predictions are blurry, and $p(c|x)$ is trained on clean images, making it difficult to obtain the accurate corresponding conditions needed for training NCT. Exploring the extension of NCT to standard diffusion models would be interesting future work.
>
> > Could you please provide side-by-side visual comparisons of your method's outputs against those from other one-step controllable generation models, such as JDM?
>
> Thanks for the great suggestion. We will provide the visual samples in revision. But limited to the policy of rebuttal, we cannot provide the samples in the rebuttal stage. Sorry for this.
>
> > On the Generalizability of NCT: The experiments currently focus on structural control signals, such as Canny edges and depth maps. Have you explored the efficacy of NCT with non-structural or stylistic control signals, for instance, controlling for a specific texture or an artistic style? Could you elaborate on how the core assumption of NCT—that diffusing noise progressively decouples it from the condition—is expected to hold for features that are less spatially robust? For such controls, even minor perturbations in the noise space might lead to significant semantic deviations, potentially challenging the consistency objective.
>
> We highlight that, theoretically, using NCT to inject conditions for one-step generation does not depend on what the condition is. In particular, in Theorem 1, we show that our optimization objective can learn a conditional generator $f_{\theta,\phi}(x,c)$, and that the validity of this theorem is independent of the type of the condition $c$.
>
> Empirically, for the generation of injected image styles, we already have a visualized image based on IP-Adapter in our main paper (Figure 3). In the second row of Figure 3, we can effectively use Van Gogh's paintings as an image prompt to inject style, which can effectively infuse the model-generated images with the Van Gogh style.

---

> ### Comment · Reviewer_1vDf · 2025-08-04
>
> Thank you for your reply. I have some more questions.
>
> 1. Thank you for comparing your results to JDM. I think even if it scores similar to JDM your methods is more elegant, efficient. I think NCT has advantages than JDM. I hope authors should include the table compared to JDM and NCT, with qualitative comparison later.
>
> 2. Regarding Style-Controlled Generation
>
> For my understanding, the input is (x,c) and the author mentioned that  "we can effectively use Van Gogh's paintings as an image prompt to inject style", however, It does not mean **inject style as controllable condition**. It just processed images as  input. So, the questions about high-frequency feature, are not addressed. For example, you can condition magnitude of images as condition after  DFFT, as because magnitude are consider to be style. What I am curious is that, For my understanding, you intuition is built upon  "if you inference similar noise, you will got similar images", because it is one-step generation.
>
> Since most of my concerns are solved, I am willing to adjust my score in final decision stage.

---

> > ### Author Response · Authors · 2025-08-05
> >
> > Thanks for the acknowledgment of our work and for the willing to raise the score!
> >
> > > Thank you for comparing your results to JDM. I think even if it scores similar to JDM your methods is more elegant, efficient. I think NCT has advantages than JDM. I hope authors should include the table compared to JDM and NCT, with qualitative comparison later.
> >
> > Thanks for your appreciation of our work. We will add the table and qualitative comparison in our revision.
> >
> > > For my understanding, the input is (x,c) and the author mentioned that "we can effectively use Van Gogh's paintings as an image prompt to inject style", however, It does not mean inject style as controllable condition. It just processed images as input. So, the questions about high-frequency feature, are not addressed. For example, you can condition magnitude of images as condition after DFFT, as because magnitude are consider to be style. What I am curious is that, For my understanding, you intuition is built upon 'if you inference similar noise, you will got similar images', because it is one-step generation.
> >
> > We clarify that the input for our method of the IP-Adapter's case is also $(\epsilon, c)$.
> > In this case, we would like to further clarify that the input condition for style control is the informative features obtained by frozen CLIP's image encoder rather than direct image input.
> > Moreover, *we can apply IP-Adapter for injecting styles rather than content, by adjusting the injecting layers as studied by InstantStyle [a] which is used to generate our Figure 3.*
> >
> > We agree that exploring the direct injection of high-frequency features (e.g., obtained by DFFT) can be interesting future work, however, it needs extra work in designing the specific adapter for injecting. We will explore this interesting setting in the future.
> >
> > Great observation on our intuition. The intuition behind this method indeed is "similar noises produce similar images", thus also produces similar conditions corresponding to images. Hence we believe our noise consistency training can learns the conditions.
> >
> > [a] InstantStyle: Free Lunch towards Style-Preserving in Text-to-Image Generation

---

### Official Review · Reviewer_MrHE · 2025-07-03

**Clarity:** 2
**Significance:** 3
**Originality:** 3
**Rating:** 4
**Confidence:** 3

**Summary:**

This paper introduces Noise Consistency Training (NCT), a novel method for integrating new control signals into pre-trained one-step generative models. The core contribution is to achieve this without the original training images or retraining the base diffusion model. The methodology involves training an adapter module with two losses: a noise consistency loss, which enforces that the generator's outputs remain consistent for noise vectors sampled along a diffusion trajectory, and a boundary loss, which acts as a regularizer to ensure the adapted model's output distribution does not collapse. By enforcing this consistency objective, NCT theoretically works to minimize the distributional distance between the adapted generator and the target conditional distribution. The authors demonstrate through extensive experiments that this approach achieves state-of-the-art controllable generation in a single forward pass, claiming superiority in both image quality and computational efficiency over existing multi-step and full distillation-based methods.

**Questions:**

1. I suggest adding something indicating noise injection in Figure 1. Its current version is not very intuitive to understand, and all 4 noises look exactly the same while there are 3 levels.
2. Is the primal-dual algorithm really necessary? Have you tried sweeping different values of the dual weight? Related question: What is the computational overhead of the primal-dual algorithm, compared to just primal ($\lambda$ is a hyperparameter)?

**Ethical Concerns:**

["NO or VERY MINOR ethics concerns only"]

**Final Justification:**

I do not have more questions, and I think it is an interesting work. Since I am not an expert on training diffusion models to obey control, I give a borderline assessment with low confidence.

**Limitations:**

Yes.

**Paper Formatting Concerns:**

No.

**Quality:**

2

**Strengths And Weaknesses:**

**Strength**
1. The proposed method, to the best of my knowledge, is novel, and it provides a practical, data-efficient, and modular solution to an increasingly relevant problem.
2. The idea of "noising" and "denoising" the initial Gaussian noise is interesting.
3. The experimental results seem quite promising, and the figures are all well-made.

**Weakness**
1. I feel the writing of the paper can be improved. For example, in the main text of the paper, it is not clearly explained the notion of $p(c|x)$, which throws me off in the experiment section. I was confused about how the model learns the new control signals when the alignment is with the original model. I hope I understand it correctly, and we assume we have online access to turning samples into the underlying conditions. This brings me to point 2.
2. What if the control signal is data-driven? Specifically, if we assume that we have access to paired data of conditions and samples but no model outputting conditions given samples, we could still do ControlNet then distillation, but the proposed method here will not work, correct?
3. Technically, this is not a weakness, but the need for an additional boundary loss may be concerning.

---

> ### Author Rebuttal · Authors · 2025-07-31
>
> We sincerely thank you for your time and effort in reviewing our paper. We address each comment below.
>
> > It is not clearly explained the notion of $p(c|x)$. Do we assume we have online access to turning samples into the underlying condition?
>
> The $p(c|x)$ is indeed for sampling the underlying condition $c$ regarding $x$. This is for constructing the triplet $(\epsilon,x,c)$ for conducting NCT's training, where the diffusing process in noise space is performed on $(\epsilon,c)$ and the boundary is defined by $(\epsilon,x)$.
>
> > What if the control signal is data-driven? Specifically, if we assume that we have access to paired data of conditions and samples but no model outputting conditions given samples, we could still do ControlNet then distillation, but the proposed method here will not work, correct?
>
> Thanks for the great question.
> We note that the training of NCT requires triplets $(\epsilon,x,c)$.
> Hence, our method can be extended to the data-driven case by a minor modification: train an encoder to invert $x$ to noise $\epsilon$, then formulate the triplet $(\epsilon,x,c)$ for performing NCT's training. We note that training the encoder may be expensive and difficult for diffusion models, but it is easy for a one-step generator. In particular, the encoder can be trained by minimizing reconstruction loss: $ \mathrm{Min} _ { \mathrm{Enc} \in \mathcal{F}} | | f_\theta(\mathrm{Enc}(x)) - x | | _2^2$.
> We train an encoder initialized from DMs in just 5,000 iterations. We report NCT trained by data-driven control signal as follows:
>
> | Method                      | FID↓  | Con.↓ |
> | --------------------------- | ----- | ----- |
> | Ours w/ data-driven signals | 14.03 | 0.108 |
> | Ours                        | 13.67 | 0.110 |
>
> It can be seen that the data-driven NCT can also achieve promising results.
>
> > I suggest adding something indicating noise injection in Figure 1. Its current version is not very intuitive to understand, and all 4 noises look exactly the same while there are 3 levels.
>
> Great suggestion. We will improve the Figure 1 in the revision. Specifically, we will add some structural/semantic features to the noise to enhance readability, rather than faithfully rendering standard Gaussian noise.
>
> > Is the primal-dual algorithm really necessary? Have you tried sweeping different values of the dual weight? Related question: What is the computational overhead of the primal-dual algorithm, compared to just primal ( $\lambda$ is a hyperparameter)?
>
> Without primal-dual can also work reasonably well as shown in Table 3. The major advantage of primal-dual is that it can enable the dynamic tuning of $\lambda$. Notably, it can allow $\lambda$ to be zero, when the boundary is nearly satisfied. This can not be achieved by any fixed $\lambda$ value (except zero).
>
> The computational overhead is negligible, since we just need to compute $\lambda_{t+1} = \mathrm{max}[{\lambda_t + \eta (L_{con} - \xi),0}] $ without additional network forward and backward, which is highly efficient.

---

> > ### Comment · Reviewer_MrHE · 2025-08-07
> >
> > I thank the authors for their detailed response, and I do not have any further questions.

---

> > > ### Author Response · Authors · 2025-08-08
> > >
> > > Thanks for your reply. We are glad that our response has answered your questions. Thanks again for your time and effort in reviewing our paper.

---

### Author Response · Authors · 2025-08-09
**Response Summary**

We thank the area chair and all reviewers for your time, insightful suggestions, and valuable comments. Your suggestions have been invaluable in refining our work, and we deeply appreciate the time and effort you dedicated to reviewing our paper. We have carefully addressed all points in our response.

We are encouraged by the reviewers’ positive feedback on various aspects of our work:

- Reviewer MrHE: The proposed method, to the best of my knowledge, is novel, and it provides a practical, data-efficient, and modular solution to an increasingly relevant problem.

- Reviewer 1vDf: This approach offers significant academic contributions by presenting a new avenue for adding controls directly to pre-trained models, thereby avoiding expensive retraining or distillation processes.

- Reviewer X1xZ: The paper introduces Noise Consistency Training (NCT), a conceptually simple yet novel approach that enables controllable image generation using a one-step generator.

- Reviewer 92Eo: The paper tackles a highly relevant and practical challenge in AIGC: efficiently adapting powerful one-step generators to diverse control signals without costly retraining or distillation.

We are also pleased that our clarifications and additional experiments have been well-received:

- Reviewer MrHE: I thank the authors for their detailed response, and I do not have any further questions.

- Reviewer 1vDf: Since most of my concerns are solved, I am willing to adjust my score in final decision stage.

- Reviewer 92Eo: Thanks for the detailed reply. It resolves some of my concerns

**Additional Experiments and Improvements**

We also sincerely thank the reviewers for their valuable suggestions, which helped us identify areas for improvement. In response to their feedback, we have made several additional experiments to further strengthen our work:

- **Extension to Data-Driven Control Signals** (Reviewer MrHE): We demonstrated that NCT can be effectively extended to scenarios where only paired data is available, showcasing its flexibility.

- **Comparison with JDM on Image-Prompted Generation** (Reviewer 1vDf): As suggested, we have included a direct comparison with JDM for the image-prompted generation task. The results show that NCT achieves competitive performance without relying on an additional distillation loss.

- **Strengthened Ablation Studies** (Reviewer 1vDf, 92Eo): We have expanded our ablation studies to include a "direct denoising" baseline. The results confirm that this simpler approach yields significantly worse FID scores due to blurry outputs, underscoring the critical role of our proposed noise consistency loss.

- **Empirical Variance Analysis** (Reviewer X1xZ, 92Eo): To provide concrete evidence for our claim that noise consistency training has lower variance, we conducted an empirical analysis. The results quantitatively demonstrate that our objective's variance is substantially lower than that of direct denoising.

- **Applicability to Non-Structural Controls** (Reviewer 1vDf): We clarified that NCT is effective for non-structural and stylistic controls, as empirically validated by our experiments with IP-Adapter (Figure 3), where artistic styles are successfully injected from reference images.

We believe our additional experiments and clarifications have satisfactorily addressed the reviewers' concerns. We highlight that ***our NCT is the first algorithm that enables adding controls to the one-step generator without relying on extra diffusion distillation or retraining the base diffusion model***. We are confident that the paper now presents a more robust and well-supported contribution.
We thank all reviewers again for their constructive feedback and for helping us improve the quality of our work.

---

### Decision · Program_Chairs · 2025-09-17

**Decision:**

Accept (poster)

**Comment:**

This paper introduces Noise Consistency Training (NCT) -- a method to incorporate additional control to a pre-trained one step diffusion generator using an additional trainable model such as ControlNet. The core idea is to optimize a consistency type loss on the one step trainable model outputs on two noisy versions of the initial coupled noise $z$ and condition $c$.

Reviewers appreciated the method’s novelty to a relevant and timely problem, the practical and interesting solution with "noising the initial noise" approach, and the solid results especially for a one step method. Reviewers’ concerns included exposition issues; questions about the availability of samples $p(c|x)$, which the authors provided a remedy in rebuttal by inversion of the generation to find noise matching an image and condition pair; and concerns about experiments including baseline selection and limited score, which the authors also addressed during rebuttal to the satisfaction of the relevant reviewer.
The negative reviewers maintained their weak-reject scores after discussion but lowered their confidence with one of the reviewers explicitly not objecting to acceptance. The second negative reviewer main concern was writing quality which didn't seem to be a strong consensus concern. Overall, a weak accept recommendation here.